# Incentivized Exploration with Stochastic Covariates: A Two-Stage Mechanism Design for Recommender System

Yuantong Li [1]   Guang Cheng [2]   Xiaowu Dai [2]

## Abstract

Recommender systems play a crucial role in internet economies by connecting users with relevant products. However, designing effective recommender systems faces the key challenges: the *exploration-exploitation* tradeoff in securing *incentive* to explore new products against user's self-interested preferences. While prior work addresses Bayesian Incentive Compatibility (BIC) in fixed-design linear bandits (Sellke & Slivkins, 2023), we tackle the challenge of stochastic user covariates sampled online. Unlike standard black-box reductions (Mansour et al., 2020), our two-stage framework exploits the linear reward structure to achieve sublinear regret while satisfying incentive constraints. To address it, we propose a two-stage algorithm that integrates incentivized exploration with *any efficient plug-in offline learning algorithms*. In the first stage, it explores products while maintaining incentive compatibility to gather optimal samples. The second stage employs *inverse proportional gap sampling strategy* (IPGS) integrated with any efficient learning methods to secure sublinear regret. Theoretically, we prove that algorithm `RCB` achieves $\tilde{O}(\sqrt{KdT})$ regret and simultaneously satisfies incentive constraints, and discovers the tradeoff between incentive budget and regret, validating in experiments. We demonstrate `RCB`'s strong incentive gain, sublinear regret, and robustness through a real application on personalized warfarin dosing and simulations.

## 1. Introduction

In the current era of the internet economy, recommender systems have been widely adopted across various domains such as advertising, consumer goods, music, videos, news, job markets, and travel routes (Koren et al., 2009; Li et al., 2010; Covington et al., 2016; Wang et al., 2017; Zheng et al., 2018; McInerney et al., 2018; Naumov et al., 2019; Lewis et al., 2020; Bao et al., 2023; Zhai et al., 2024; Jeon et al., 2024). Modern recommendation markets typically involve three key stakeholders: products, users, and the platform or called principal. The platform collects and analyzes user data to enhance future distribution services and to respond effectively and promptly. In these personalized recommendation markets, the platform fulfills a dual role: recommending the best product to users (exploitation role) and experimenting with lesser-known products to gather more information to enlarge a high quality of products pool (exploration role). Because users are self-interested and heterogeneous, they rarely voluntarily explore products with low prior estimates based on the common knowledge (Dai et al., 2024). Without intervention, these products cannot collect the data required to overcome the cold-start phase and achieve broad adoption. While exploration generates valuable information for the platform and future users, self-interested users often find it costly. Consequently, feedback remains sparse, creating a need for mechanisms that strategically incentivize efficient data collection.

The fundamental challenge lies in the tension between the platform's long-term learning objectives and the users' immediate self-interest. To summarize, the platform faces a mechanism design problem with two coupled objectives:

- *classic exploration-exploitation tradeoff*: How to design a recommendation policy that maximizes cumulative rewards by balancing the acquisition of new information against the utilization of known preferences.

- *dynamic incentive compatibility*: How to leverage information asymmetry to incentivize heterogeneous users to accept exploratory recommendations, thereby preventing the systemic bias and sub-optimality caused by myopic behavior from users.

The multi-armed bandit (MAB) framework for incentivized exploration was established by (Kremer et al., 2014; Mansour et al., 2020). While foundational, these early models typically assume independent priors. Subsequent work extended this to correlated priors via Thompson Sampling

[1]Meta Platforms Inc, New York, NY, United States [2]Department of Statistics and Data Science, University of California, Los Angeles, CA, United States. Correspondence to: Yuantong Li <yuantongli@meta.com>, Xiaowu Dai <daix@ucla.edu>.

*Proceedings of the $43^{rd}$ International Conference on Machine Learning*, Seoul, South Korea. PMLR 306, 2026. Copyright 2026 by the author(s).

*Table 1.* Comprehensive comparison of `RCB` with prior BIC literature.

| Algorithm | Problem Setting | Context Model | Algorithmic Mechanism | Key Gap Filled |
|---|---|---|---|---|
| Kremer et al. (2014) | Multi-Armed Bandit | No Context | Optimal Policy Design | Initiated BIC exploration for MAB |
| Mansour et al. (2020) | Multi-Armed Bandit | None or Black-box reduction (ignores linear structure) | MAB Greedy + Hidden Exploration (Phases) | Establishes BIC for general MAB |
| Sellke (2023) | Linear Contextual Bandit | Fixed Design (Features are static/owned by products) | Ridge Regression + Phased Thompson Sampling | Analyzes linear rewards under fixed contexts |
| **RCB (Ours)** | **Linear Contextual Bandit** | **Stochastic Covariates (User features sampled online)** | **Two-Stage (Cold Start + IPGS Gap Sampling)** | **Handles dynamic user contexts where best arm changes per round** |

(Hu et al., 2022; Sellke, 2023). Hu et al. (2022) focus on combinatorial semi-bandits without personalized contexts. Furthermore, while Sellke (2023) and Kalvit et al. (2024) address linear contexts, they operate under a fixed design assumption (static arm features). In contrast, our work addresses *stochastic user covariates* sampled online. This dynamic setting, where the optimal arm varies per user, precludes the fixed-design analyses or generic black-box reductions used in prior work, Table 1.

In this paper, we first formalize those challenges into a *dynamic Bayesian incentive compatibility* (DBIC) problem. That is, the platform can not only provide personalized recommendations with dynamic user context, but also need to consider to predefine the optimal exploration budget for cold start contents.

We propose the *recommendation contextual bandit* algorithm (RCB, Algo 1) to solve this problem, which is composed of a two-stage design's algorithm. In the first stage, the platform explores all available products with precomputed optimal sample sizes, collecting the necessary data for each content to prepare for the subsequent stage. The second stage employs an *inverse proportional gap sampling bandit* strategy integrated with any efficient plug-in offline machine learning method to secure DBIC constraint and sublinear regret. Our algorithm simultaneously secure sublinear regret and achieve DBIC constraint over the whole process. Our main contributions can be delineated into three parts:

1. We formalize the DBIC in §2. Then we introduce the RCB algorithm, a two-stage framework that fundamentally departs from the posterior sampling methods dominant in prior art. Instead of relying on specific posterior forms (such as Gaussian or Beta distributions), RCB employs a novel **inverse proportional gap sampling** strategy. This design enables a **modular architecture**: RCB can seamlessly integrate *any efficient offline learning oracle* to estimate rewards while strictly maintaining the DBIC.

2. Theoretically, our contributions are two-fold:

- *regret bounds:* We establish that RCB achieves a regret bound of $\tilde{\mathcal{O}}(\sqrt{KdT})$, scaling efficiently with the feature dimension $d$ and horizon $T$, matching standard contextual bandit rates while satisfying the strict incentive constraints.

- *price of incentives:* We formally quantify the *price of incentivizing exploration* (Sellke & Slivkins, 2023) in the setting of stochastic covariates. We derive the minimal cold-start sample complexity $N(\epsilon)$ required to initialize the learning process. This reveals the precise trade-off between the user's incentive budget $\epsilon$ and the platform's exploration efficiency, proving that a larger budget constraint ($\epsilon$) significantly accelerates the transition to the sublinear regret regime.

3. Lastly, we empirically validate the effectiveness of RCB through its performance in terms of incentive gain and sublinear regret, and its robustness across various environmental and hyperparameter settings. Additionally, we apply our algorithm to a case study, personalized warfarin dose allocation, and compare it with other methods to demonstrate its efficacy.

**Notations.** We denote $[N] = [1, 2, ..., N]$ where $N$ is a positive integer. Define $x \in \mathbb{R}^d$ be a $d$-dimensional random vector. The capital $X \in \mathbb{R}^{d \times d}$ represents a $d \times d$ real-valued matrix. Let $I_d$ represent a $d \times d$ diagonal identity matrix. We use $\mathcal{O}(\cdot)$ to denote the asymptotic complexity. We denote $T$ as the time horizon.

## 2. Problem Formulation

Assume a sequence of streaming users $(T)$ arrive to the platform to receive service (music, video, etc,.), each user $p_t$ with features $x_t \in \mathcal{X} \subset \mathbb{R}^d$. In this platform, there is a set of products $\mathcal{A}$ ($|\mathcal{A}| = K$), where each product can be represented as an unknown parameter $\beta \in \mathbb{R}^d$. Then the platform provides the personalized recommendation to user with the following protocol:

1. The platform provides a personalized recommendation to this user with a group of products and denote the best

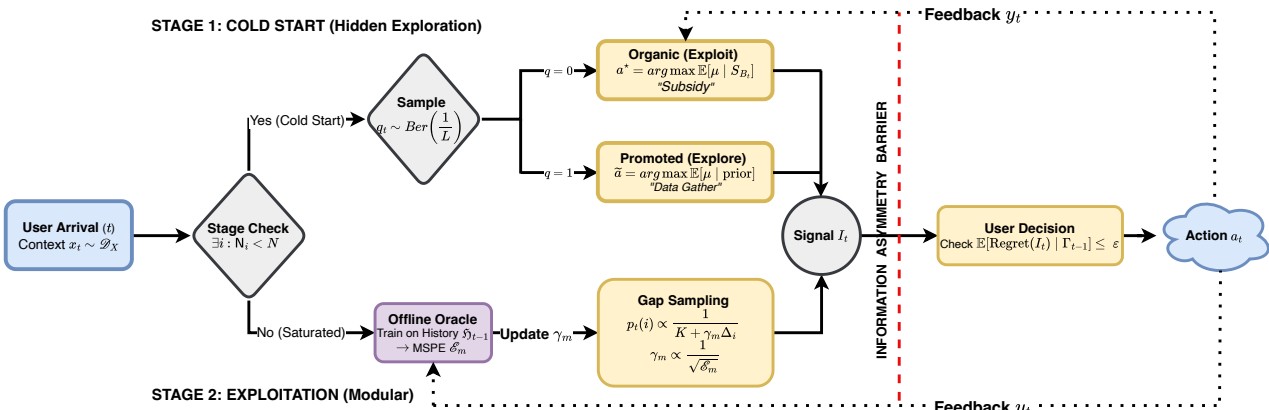

*Figure 1.* RCB Algorithm

arm as $I_t$.

2. User chooses an action $a_t \in \mathcal{A}$, $a_t$ might be not be equal to $I_t$, and then the platform receives noisy feedback $y_t(a_t) \in [0,1]$. Here we assume the feedback $y_t(a_t)$ following the linear model

$$y_t(a_t) = \mu(x_t, a_t) + \eta_{t,a_t}, \quad (1)$$

where $\mu(x_t, a_t) = x_t^\mathsf{T} \beta_{a_t}$ is the true mean reward, [1] and $\{\eta_{t,a_t}\}_{t \geq 1}$ are $\sigma$-subgaussian random variables and independent of the covariates $\{x_t\}_{t \geq 1}$.[2]

Besides, for notation simplicity, we denote $y_t \in [0,1]^K$ as all potential rewards in vectorized format. Similarly, $\mu(x_t) \in [0,1]^K$ as the all potential true rewards, and $\eta_t \in \mathbb{R}^K$ as the all potential vector noise. Without loss of generality, we assume $\mathcal{X}$ and $\beta$ are bounded. In addition, in this formulation, we have two random sources: the covariate vector $x_t$ and noise $\eta_t$, different from the fixed design $\{x_t\}_{t \geq 1}$ (Lattimore & Szepesvári, 2020). Here we assumed a shared prior belief $\mathcal{P}_0$ that factorizes over products as $\mathcal{P}_0 = \mathcal{P}_{1,0} \times \ldots \times \mathcal{P}_{K,0}$. Each product parameter $\beta_i \sim \mathcal{P}_{i,0}$ is drawn from this prior with mean $\beta_{i,0} = \mathbb{E}[\beta_i]$ and covariance $\Sigma_{i,0}$. Given the stochastic covariate $x_t$, we define the prior mean reward for product $i$ as $\mu_0(x_t, i) = \mathbb{E}[\mu(x_t, i)]$.

A fundamental distinction between this protocol and standard sequential decision-making (Sutton & Barto, 2018; Lattimore & Szepesvári, 2020) is the potential for user non-compliance. While the platform recommends the best arm $I_t$, the self-interested user $p_t$ selects the final action $a_t$ based on the recommendation and his posterior beliefs together. Consequently, $a_t$ may differ from $I_t$, and the platform observes feedback only for the user-selected arm $a_t$. This

contrasts with standard bandits, where promoted content $I_t$ can always collect feedback.

While the platform aims to recommend an arm $I_t$ that maximizes long-term social welfare (exploration), self-interested users are myopic: they seek to maximize their immediate expected reward conditional on their own feelings (priors) and information revealed by the platform. Consequently, this user may reject a recommendation if it appears suboptimal relative to their private belief. To align these objectives, we leverage information asymmetry to achieve better exploration and exploitation, and long term social welfare: the platform observes the full history of rewards, while the user observes only the recommended actions and the prior.

However, we find that with prior information and recommended products, a rational and myopic user would follow $I_t$, if the platform satisfies the $\epsilon$-DBIC constraint, formally defined as follows:

**Definition 1** ($\epsilon$-DBIC). Denote the public history under the assumption that previous users have followed recommendations as $\Gamma_{t-1} = \{I_s = a_s : s \in [t-1]\} \cup \mathcal{P}_0$. Given an *incentive budget* $\epsilon \geq 0$, a recommendation algorithm is $\epsilon$-dynamic Bayesian incentive-compatible (DBIC) if

$$\mathbb{E}[\mu(x_t, i) - \mu(x_t, j)|I_t = i, \Gamma_{t-1}] \geq -\epsilon, \\ \forall t \in [T], i, j \in [K], i \neq j. \quad (2)$$

If $\epsilon = 0$, we call it dynamic Bayesian incentive-compatible (DBIC). For brevity, we use the terms DBIC and $\epsilon$-DBIC interchangeably throughout this paper, unless the distinction is explicitly emphasized.

This definition establishes a *compliance condition*: a rational, myopic user will follow the recommendation $I_t$ provided that the expected posterior utility of $I_t$ is not outperformed by any alternative product $j$ by more than the incentive budget $\epsilon$. Each user has a personalized and unknown incentive budget, which functions like a credit balance on

---

[1] The discussion of the nonlinear reward format is available in Appendix §G.

[2] if $\mathbb{E}[e^{t\eta}] \leq e^{\sigma^2 t^2/2}$ for every $t \in \mathbb{R}$. For non-subgaussian r.v.s, we can refer the sub-exponential r.v. techniques (Jia et al., 2022)

the platform. Specifically, the user selects the product maximizing their posterior expected reward given the history $\Gamma_{t-1}$ and the recommended product $I_t$; the $\epsilon$-DBIC constraint ensures that the recommended product effectively satisfy this constraint, incentivizing the user to adhere to the platform's best suggestion $I_t$.

**Myopic User Model.** The myopic user assumption is standard in the incentivized exploration literature (Kremer et al., 2014; Mansour et al., 2020; Sellke, 2023). It captures the empirically observed "bounded rationality" where users evaluate recommendations based on their current belief and conditional information about immediate rewards. By incorporating an $\epsilon$-budget, we model how users strategically choose content even if it is not their immediate myopic best, acting as a credit balance of trust.

From the perspective of the platform, the goal is to map the available information to a recommendation arm. At each round $t$, the platform observes the stochastic covariate $x_t$ and the full private history $\mathfrak{S}_{1:t-1} = \sigma(\{(x_t, y_t, a_t)\}_{1:t})$. The platform designs a sequential policy $\pi = \{\pi_t(\cdot)\}_{t \geq 1}$, where $\pi_t(\cdot | \mathfrak{S}_{1:t-1}) : \mathcal{X} \to \mathcal{A}$ maps the feature and history to a recommended arm. The objective is to maximize the cumulative expected reward over the horizon $T$, subject to the strict constraint that every recommendation must satisfy $\epsilon$-DBIC for each user. We evaluate the performance of the policy $\pi$ using Bayesian Regret, defined as follows:

$$\text{Reg}_{[T]}(\pi) = \sum_{t=1}^{T} \mathbb{E}\big[\mu(x_t, \pi_t^*(x_t)) - \mu(x_t, \pi_t(x_t))\big]$$
(3)

where $\pi_t^*(x_t)$ is the posterior optimal arm given all information up to $t-1$. Finally, we summarize the key challenge in the DBIC:

---
**Key challenge:**

The principal challenge lies in the misalignment between the platform's long-term objective and the users' immediate self-interest. While the platform seeks to maximize cumulative rewards through exploration, users with **dynamic, context-dependent priors** are myopic: they reject recommendations that deviate from their greedy preference unless the expected loss is bounded by an incentive budget $\epsilon$. Consequently, the platform must design a mechanism that **simultaneously** satisfies the $\epsilon$-DBIC and efficiently gathers data for cold-start products to secure sublinear regret.

---

## 3. Algorithms

To minimize regret while strictly adhering to $\epsilon$-DBIC, we propose the *Recommendation Context Bandit* algorithm (RCB, Algo 1). This two-stage algorithm begins with a *cold start* phase that collects the minimal optimal sample

size required to satisfy $\epsilon$-DBIC constraint. Later, this transitions to an *exploitation* phase, which employs a modular *inverse proportional gap sampling* (IPGS) strategy compatible with any efficient offline learning oracle. By dynamically calibrating the $\epsilon$-budget via sequential spread parameters $\{\gamma_m\}_m$, RCB simultaneously secures sublinear regret and satisfies $\epsilon$-DBIC.

### 3.1. Cold Start Stage
The objective of the cold start stage is to collect a minimum sample size $\mathsf{N}(\epsilon)$ for each arm to ensure the stability of the subsequent Exploitation stage, while strictly maintaining the $\epsilon$-DBIC in cold start stage. This stage relies on a calibrated *exploration probability $L$*, which determines the frequency of exploratory recommendations relative to exploitative ones to satisfy the incentive constraint.

**Notation**: Let $N_i(t)$ denote the number of times arm $i$ has been pulled up to time $t$. We define the set of "saturated" arms that have met the sample requirement as $B_t = \{i \mid N_i(t) = \mathsf{N}\}$. The historical data for arm $i$ is denoted by $S_i$, comprising the set of observed covariates and rewards.

The process proceeds in two phases to handle stochastic user covariates:

**(1) most popular arm's sample collection (MPASC).** The platform initially recommends the arm with the highest context-dependent prior mean reward. This continues until at least one arm enters $B_t$, establishing a "safe" baseline for the user population.

**(2) rest arm sample's collection (RASC).** Once a safe arm exists ($B_t \neq \emptyset$), the platform leverages information asymmetry to explore the remaining arms. It recommends an under-sampled arm with probability $1/L$ and exploits the safe arm with probability $1 - 1/L$. The parameter $L$ is strategically calculated to ensure the expected utility loss of exploration is fully subsidized by the high utility of the safe arm, thereby satisfying the $\epsilon$-DBIC constraint.

**a) promoted recommendation (Exploration).** When yields $q_t = 1$, the platform enters an exploration mode. It identifies the set of under-sampled arms, $[K]/B_t$, which have not yet met the sample threshold $\mathsf{N}(\epsilon)$. To maximize the likelihood of user compliance within this set, the platform recommends the *promoted arm $\widetilde{a}_t$* with the highest context-dependent prior mean:

$$\widetilde{a}_t = \underset{i \in [K]/B_t}{\arg\max} \mathbb{E}[\mu(x_t, i)].$$
(4)

Upon the user's acceptance and the observation of reward $y_{t,\widetilde{a}_t}$, the platform updates the sufficient statistics for this arm: $N_{\widetilde{a}_t}(t) \leftarrow N_{\widetilde{a}_t}(t-1) + 1, S_{\widetilde{a}_t} \leftarrow S_{\widetilde{a}_t} \cup (x_t, y_{t,\widetilde{a}_t})$. Once the count $\widetilde{a}_t$ reaches $\mathsf{N}$, the arm is deemed "saturated" and added to the set $B_t$.

**b) Organic Recommendation.** When $q_t = 0$, the platform prioritizes incentive alignment by recommending the *organic arm* $a_t^*$. This arm is selected to maximize the expected reward conditional on the currently trusted history $S_{B_t}$:

$$a_t^* = \underset{i \in [K]}{\operatorname{argmax}} \mathbb{E}[\mu(x_t, i) | S_{B_t}]. \tag{5}$$

Note that the conditional expectation $\mathbb{E}[\cdot | S_{B_t}]$ effectively utilizes the *posterior mean* for saturated arms ($i \in B_t$) and the *prior mean* for under-sampled arms ($i \notin B_t$). Crucially, while the agent $p_t$ generates a reward $y_{t,a_t^*}$, the platform does *not* increment the sample counts (N) or update the exploration history ($S$) for this organic recommendation. This ensures that the saturated rounds serve purely to "subsidize" the exploration risk, maintaining the $\epsilon$-DBIC without biasing the quick stop for the cold-start phase.

During the cold start stage, saturated arms are recommended purely as "organic" choices to subsidize the risk of exploring under-sampled arms without losing trust from myopic users. However, by not counting these pulls in the exploration history, we prevent over-counting. This ensures that the confidence sets used in Stage 2 reflect only the "useful" samples that actively contribute to reducing uncertainty and triggering epoch transitions.

### 3.2. Exploitation Stage
Following the cold start stage, the platform possesses sufficient data $S$ (with $N(\epsilon)$ samples per arm) to initialize the learning process. The objective in the exploitation stage is to maximize cumulative rewards by recommending arms with high posterior means, subject to the strict $\epsilon$-DBIC constraint. The core algorithmic challenge lies in dynamically calibrating the exploration rate to ensure that the expected loss of any recommendation never exceeds the incentive budget $\epsilon$.

To address this, RCB employs a **doubling epoch schedule** to iteratively refine the exploration-exploitation balance. The timeline is partitioned into epochs $\mathcal{T}_m = \{t \in [2^{m-1}, 2^m) \mid m \geq m_0\}$, where the exploitation phase begins at epoch $m_0 = \lceil 2 + \log_2 N \rceil$. At the beginning of each epoch $m$, the algorithm utilizes all historical data from previous epoch $W_{\mathcal{T}_{1:m-1}}$ to update a *spread parameter* $\gamma_m$. This parameter governs the "inverse proportional gap sampling" (IPGS) distribution (defined in Algorithm 2), effectively tightening the exploration radius as the offline oracle's prediction error decreases. At epoch $m \in [m_0, m_1]$, the platform employs the offline oracle trained on $W_{\mathcal{T}_{m-1}}$ to compute posterior estimators $\widehat{\beta}_i$ and predictive rewards $\widehat{\mu}_t(x_t, i) = x_t^\top \widehat{\beta}_i$. Identifying the predicted best arm as $b_t = \operatorname{argmax}_{i \in [K]} \widehat{\mu}_t(x_t, i)$, the algorithm samples recommendations $a_t$ via IPGS:

$$p_t(i) = \begin{cases} 1 - \sum_{j \neq b_t} p_t(j), & \text{if } i = b_t, \\ \frac{1}{K + \gamma_m(\widehat{\mu}_t(x_t, b_t) - \widehat{\mu}_t(x_t, i))}, & \text{if } i \neq b_t. \end{cases} \tag{6}$$

The spread parameter $\gamma_m = 4\sqrt{K/\mathcal{E}_{\mathcal{F},\delta}(|\mathcal{T}_{m-1}|)}$ scales inversely with the square root of the offline learner's Mean Squared Prediction Error (MSPE). Consequently, as the MSPE diminishes over time, $\gamma_m$ increases, dynamically concentrating recommendations on the predicted best arm to maintain incentive compatibility (see Remark 1 for detailed discussion). We formally define the offline oracle's generalization error bound, $\mathcal{E}_{\mathcal{F},\delta}(n)$, as follows:

**Definition 2.** Let $p$ be an arbitrary action selection kernel. Given a sample size of $n$ data of the format $(x_i, a_i, y_{i,a_i})$, which are i.i.d. according to $(x_i, y_i) \sim \mathcal{D}, a_i \sim p(\cdot | x_i)$, the offline learning algorithm $\mathtt{Off}_\mathcal{F}$ based on the data and a general function class $\mathcal{F}$ returns a predictor $\widehat{\mu}_t(x, a) : \mathcal{X} \times \mathcal{A} \to \mathbb{R}$. For any $\delta > 0$, with probability at least $1 - \delta$, we have $\mathbb{E}_{x \sim \mathcal{P}_X, a \sim p(\cdot | x)}[\widehat{\mu}_t(x, a) - \mu(x_t, a_t)]^2 \leq \mathcal{E}_{\mathcal{F},\delta}(n)$.

**Complexity Analysis.** The complexity are from sample complexity (data requirements) and the computational runtime:

*Cold Start Stage:* The primary cost is the sample complexity required to satisfy the sufficient data for the offline oracle. To collect $N(\epsilon)$ samples for each of the $K$ arms with an exploration probability of $1/L$, the expected duration of this stage is $\mathcal{O}(KLN)$. The per-round computational overhead is negligible, requiring only the sorting of prior means.

*Exploitation Stage:* The computational cost is dominated by the training of the plug-in offline oracle. Our framework is modular: For *parametric methods* (e.g., Ridge Regression), achieving a target generalization error $\epsilon'$ typically requires a sample size of $\tilde{\mathcal{O}}(Kd/\epsilon')$, with a training runtime scaling with matrix inversion (e.g., $\mathcal{O}(d^3)$); For *non-parametric methods* (e.g., kernel methods or neural networks), the sample requirement generally scales as $\tilde{\mathcal{O}}(K/\epsilon'^2)$ or higher, depending on the hypothesis class complexity.

## 4. Theory
### 4.1. Regularity Conditions
In order to satisfy the DBIC, we list two general assumptions over the prior distribution (Mansour et al., 2020; Sellke & Slivkins, 2023).

**Assumption 1** (Sufficient Exploration / Non-Degeneracy)**.** *Let $B_t = \{j \in [K] \mid N_j(t) \geq N\}$ denote the set of "saturated" arms for which the platform has collected sufficient samples, and let $S_{B_t}$ denote the history associated with these arms. For any under-sampled arm $i \in [K] \setminus B_t$ (where $N_i(t) = 0$), we define the **Prior-Posterior Gap**, $G_t(i)$, as the difference between the prior expected reward of arm $i$ and the posterior expected reward of the current organic choice $j \in B_t$:*

$$G_t(i) = \min_{j \in B_t} \left( \mathbb{E}[\mu(x_t, i)] - \mathbb{E}[\mu(x_t, j) \mid S_{B_t}] \right).$$

*We assume there exist time-independent problem-dependent*

constants $N_{P0}, \tau_{P0}, \rho_{P0} > 0$ such that for all $N \geq N_{P0}$ and any $i \in [K] \setminus B_t$, the following probability bound holds:

$$\mathbb{P}\left(G_t(i) \geq \tau_{P0}\right) \geq \rho_{P0}.$$

This non-degeneracy assumption simply ensures that a cold-start product has a "fighting chance"—meaning its prior distribution is not entirely stochastically dominated by the saturated arms, which guarantees the incentivized exploration problem is mathematically feasible and not trivial.

Additionally, we extend this non-degeneracy to the *Exploitation Stage* (where all arms have $N$ samples). We assume that after collecting $N_{P*}$ samples, the gap between the optimal arm and the suboptimal arms is distinguishable. Specifically, the posterior gap exceeds $\tau_{P*}$ with probability at least $\rho_{P*}$. In practice, the constants $\tau_{P0}$ and $\tau_{P*}$ serve as hyperparameters regulating the exploration aggressiveness.
**Assumption 2** (Posterior Distribution Assumption). *Denote $G_t(b_t) = \min_{j \neq b_t} \mathbb{E}[\mu(x_t, b_t) - \mu(x_t, j)|S]$ as the minimum posterior gap when we have $N$ samples of each arms in the Exploitation stage. There exist a uniform time-independent posterior constants $n_{\mathcal{P}_*}, \tau_{\mathcal{P}_*}, \rho_{\mathcal{P}_*} > 0$ such that $\forall n \geq n_{\mathcal{P}_*}, i \in [K]$, then $Pr(G_t(b_t) \geq \tau_{\mathcal{P}_*}) \geq \rho_{\mathcal{P}_*}$.*

This assumption ensures that the expected reward gap between arms is bounded away from zero, meaning the optimal arm is eventually distinguishable. This is naturally satisfied in recommendation systems where user features (e.g., demographics) have bounded or normalized support.

Then we provide the regularity conditions over covariates $\mathcal{P}_X$ as follows to avoid the singularity.
**Assumption 3** (Minimum Eigenvalue of $\Sigma$). *Define the minimum eigenvalue of the covariance matrix of $X$ as $\lambda_{\min}(\Sigma) = \lambda_{\min}(\mathbb{E}_{x \sim \mathcal{P}_X}[xx^\mathsf{T}])$. There exists such a $\phi_0 > 0$ satisfying that $\lambda_{\min}(\Sigma) \geq \phi_0$.*
**Assumption 4** (Evolution of Trust). *We posit that user priors are not static but evolve to become more diffuse over time. Specifically, we assume the minimum eigenvalues of the prior covariance matrix $\Sigma_{i,0}$ increase with order $\mathcal{O}(t)$.*

This models a behavioral shift where users become more open to platform signals over time. In practice, this is a mild requirement following standard Bayesian consistency. If this assumption is violated and users remain stubborn, the algorithm remains robust, but the platform must pay a higher "Price of Incentives" by running a longer Cold Start phase (larger $N(\epsilon)$) or using a looser incentive budget.

## 4.2. Sample Complexity and Exploration Calibration
Here we provide that $N(\epsilon)$ represents the warm-start complexity required to satisfy the incentive constraints, and $L$ represents the subsidy ratio required to mask exploration.
**Theorem 1** (Sample Complexity and Exploration Calibration). *Suppose Assumptions 1–3 hold and priors are Gaus-*

sian. To guarantee that the RCB algorithm satisfies the $\epsilon$-DBIC with probability at least $\rho_{\mathcal{P}_0}\rho_{\mathcal{P}_*}$, it suffices to set the per-arm cold-start sample size $N$ and the inverse exploration probability $L$ as follows:

$$N(\epsilon) \geq \frac{(\sigma^2 d + 1)K^3}{\phi_0(\tau_{\mathcal{P}_*} + \epsilon)^2} \quad and \quad L \geq 1 + \frac{1 - \epsilon}{\tau_{\mathcal{P}_0}\rho_{\mathcal{P}_0} + \epsilon}. \quad (7)$$

*Consequently, the Exploitation stage is initialized at epoch $m_0(\epsilon) = \lceil 2 + \log_2 N(\epsilon) \rceil$.*

Theorem 1 quantifies the *Price of Incentivizing Exploration* in the presence of stochastic covariates. The bound on $N(\epsilon)$ reveals the structural dependencies required to maintain user trust during the learning process:

- *cubic in arms ($K^3$):* The sample complexity scales cubically with $K$, reflecting the difficulty of maintaining incentives when many competing arms must be simultaneously explored and compared against a dynamic best arm (Mansour et al., 2020).

- *linear in context ($d$):* The complexity is linear in the covariate dimension $d$, ensuring scalability in high-dimensional feature spaces typical of modern recommendation systems.

- *inverse quadratic in incentive budget ($(\tau + \epsilon)^{-2}$):* This term highlights the core economic trade-off: a tighter incentive budget $\epsilon$ (i.e., less tolerant users) necessitates a significantly longer cold-start phase to reduce estimator variance before the platform can safely transition to the Exploitation stage.

- *spectral dependency ($\phi_0^{-1}$):* The complexity is inversely proportional to the minimum eigenvalue of the context covariance matrix, $\phi_0$. This ensures that RCB is robust only when the user contexts are sufficiently diverse to support learning across all dimensions.

*Remark* 1 (The Decoupling of Incentives and Learning). Theorem 1 highlights a critical economic trade-off: the cold-start sample complexity $N(\epsilon)$ scales inversely with the square of the incentive budget $\epsilon$, quantifying the "price of incentives" required to initialize the system. Crucially, RCB achieves an algorithmic *decoupling* between incentive constraints and learning rates. The dependence on $\epsilon$ is encapsulated entirely within the initialization threshold $N(\epsilon)$. Once the Exploitation stage begins ($m \geq m_0$), the spread parameter $\gamma_m$ is derived solely from the offline oracle's prediction error (MSPE) and the epoch length, evolving independently of $\epsilon$. This ensures that as long as the cold-start condition is met, the natural reduction in prediction error is sufficient to satisfy the $\epsilon$-DBIC constraint dynamically.

## 4.3. Regret Upper Bound
We now establish the regret bound for RCB. The total regret is structurally decomposed into two components: the *Price*

of *Incentivizing Exploration* (incurred during the Cold Start Stage) and the standard *Learning Regret* (incurred during the Exploitation stage).

**Theorem 2** (Regret Decomposition). *Let $T_{cold} \approx m_0(\epsilon)$ denote the duration of the Cold Start stage required to satisfy the conditions in Theorem 1. Under Assumptions 1–4, for any horizon $T > T_{cold}$, with probability at least $1 - \delta$, the cumulative regret of* RCB *is bounded by:*

$$\mathcal{R}(T) \leq \underbrace{T_{cold}(\epsilon)}_{\textit{Price of Incentives}} + \underbrace{\tilde{\mathcal{O}}\left(\sqrt{Kd(T - T_{cold})}\right)}_{\textit{Learning Regret}}. \quad (8)$$

The regret bound illustrates the fundamental trade-off in incentivized exploration: *The Price of Incentivizing Exploration (Stage 1):* The first term, $T_{\text{cold}}(\epsilon)$, represents the unavoidable regret incurred to accumulate sufficient data to make the system DBIC. As established in Theorem 1, this cost scales as $\mathcal{O}(1/\epsilon^2)$. This aligns with the theoretical characterization by Sellke & Slivkins (2023), who prove that any BIC algorithm requires an initial "warm-start" phase where performance is sacrificed to build the posterior precision required to persuade myopic agents.

*Efficiency of Modular Learning (Stage 2):* The second term, scaling with $\sqrt{T}$, confirms that once the incentive constraints are stabilized, RCB achieves the standard sublinear regret rate for contextual bandits (Lattimore & Szepesvári, 2020). The dependence on $\sqrt{Kd}$ reflects the modularity of our approach: the regret is governed by the generalization error of the offline oracle (which scales with dimension $d$) and the spread parameter required to maintain DBIC (which scales with $\sqrt{K}$).

# 5. Experiments

## 5.1. Real Data

We leverage the PharmGKB dataset (5,528 patients) (Consortium, 2009) to simulate a Clinical Decision Support system for personalized warfarin dosing, aiming to mitigate the adverse effects of traditional fixed-dose strategies. In this model, the system acts as a "second opinion" for clinicians. Detailed data specifications and pre-processing steps are provided in Appendix F.4.

**Arms Construction.** Following the protocol in Bastani & Bayati (2020), we formulate the problem as a $K$-armed bandit ($K = 3$) with patient covariates. We discretize the continuous dosage space into three buckets based on clinically relevant thresholds:

- *Low (Arm 1):* $< 3$mg/day (33% of patients).

- *Medium (Arm 2):* $3 - 7$mg/day (54% of patients).

- *High (Arm 3):* $> 7$mg/day (13% of patients).

In this setting, the "Physician Assigned Dosage" (Standard of Care) corresponds to a fixed strategy of always recommending the *Medium* dose. This serves a dual purpose in our simulation: it acts as a baseline and, crucially, defines the *Agent's Prior* ($\mathcal{P}_0$). Patients requiring Low or High doses are at risk of excessive anticoagulation or thrombosis, respectively, under the Physician's prior. The challenge for RCB is to incentivize the clinician to explore the Low/High arms despite their strong prior belief in the Medium arm.

**Reward Construction.** We define a binary reward outcome: $y_t = 1$ if the recommended dosage matches the patient's true optimal arm, and $0$ otherwise. Consequently, the cumulative regret directly quantifies the total count of incorrect dosing decisions. Although the outcome is discrete, RCB utilizes a linear regression oracle to estimate the *expected* reward (i.e., the conditional probability of a correct diagnosis). This setup serves to empirically validate the algorithm's robustness to model misspecification, demonstrating its efficacy even when a linear predictor is applied to a discrete classification task.

**Ground Truth:** We estimate the true arm parameters $\beta_i$ using the linear regression with the entire dataset for specific group. Besides, we scale the optimal warfarin dosing into $[0, 1]$ with minimum dosing as 0, and maximum dosing as 1. The true mean warfarin dosage is obtained from the inner product of $\beta_i$ (based on the optimal arm) multiples the covariate of this patient. Besides, for the counterfactual arm, the true mean dosage are set to be 0.

**RCB Setup**: The total number of trials is set at $T = 5528$, with reward noise $\hat{\sigma} = 0.054$ estimated from the true optimal dosing of warfarin after scaling. To create an online decision-making scenario, we simulate the process across 10 random permutations of patient arrivals, averaging the results over these permutations. The exploration budget $\epsilon$ is varied among $[0.025, 0.035, 0.045]$. The minimum gap $\tau_{\mathcal{P}_0}$ is set at $0.005$. The prior variance is defined as $\Sigma = [0.4, 0.6, 0.8]\mathbf{I}_d$, and the prior means are $\beta_{2,0} = 0.05 \times \mathbf{I}_d$, $\beta_{1,0} = \beta_{3,0} = \mathbf{0}_d$. Further details on hyperparameters are available in §F.4.

**Evaluation Criteria.** We employ four complementary metrics to assess it:

- *cumulative regret:* We define the binary reward $y_t = 1$ if the recommended dosage matches the patient's true optimal arm, and $y_t = 0$ otherwise. Consequently, the cumulative regret $\mathcal{R}(T)$ directly quantifies the *total count of incorrect dosing decisions* over the horizon $T$.

- *$\epsilon$-DBIC Gain:* This metric tracks the instantaneous incentive compatibility of the recommendation. It measures the difference between the user's expected reward from the recommended arm $I_t$ and their best alterna-

*Table 2.* Comparison `RCB` and physician algorithm and distribution of patients.

| | | RCB Algo Assigned Dosage | | | Physician Algo Assigned Dosage | | | % of Patients |
|---|---|---|---|---|---|---|---|---|
| | | Low | Medium | High | Low | Medium | High | |
| **True Dosage** | Low | **50%** | 48% | 2% | **0%** | 100% | 0% | **27%** |
| | Medium | 14% | **84%** | 2% | 0% | **100%** | 0% | **60%** |
| | High | 2% | 93% | **5%** | 0% | 100% | **0%** | **13%** |

tive action given the history $\Gamma_{t-1}$. A value greater than $-\epsilon$ indicates that the user is incentivized to follow the recommendation.

- *fraction of incorrect decisions:* We report the error rate ($\mathcal{R}(T)/T$) to provide an interpretable metric of clinical failure. This is critical in medical settings where non-optimal arms imply specific health risks (e.g., thrombosis or hemorrhage) rather than merely lower utility.

- *weighted risk score:* To penalize "safe" heuristics (such as the Physician baseline, which always selects the 'Medium' dose), a score that accounts for the population class imbalance (Low: 27%, Medium: 60%, High: 13%). The score assigns $+1$ point for a correct decision and $-1$ point for an incorrect decision, weighted by the true dosage prevalence $p_k$, Score $= \sum_{k \in \{L,M,H\}} p_k \times (\mathbb{I}(\text{Correct}|k) - \mathbb{I}(\text{Incorrect}|k))$. This metric exposes the weakness of the Physician baseline, which achieves high accuracy on the majority class but fails on high-risk patients requiring Low/High dosages.

### 5.1.1. RESULT ANALYSIS

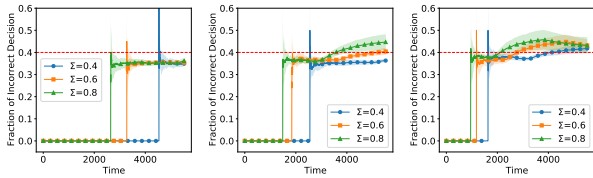

*Figure 2.* Left to right: fraction of incorrect decision under different setups of budgets ($\epsilon$) $[2.5, 3.5, 4.5] \times 10^{-2}$. Dotted line represents the lasso bandit's error rate.

In Table 2, we exhibits the `RCB`'s dosage correction ratio and physician assigned dosage correction ratio and weighted risk scores. As for cumulative regret and $\epsilon$-DBIC gain, we put it in the Appendix.

**Fraction of Incorrect Decisions.** Figure 2 presents the decision error rate, a critical metric in clinical settings where non-optimal arms carry significant health risks (e.g., thrombosis). We observe a distinct performance hierarchy governed by the incentive budget $\epsilon$ and the strength of the prior $\Sigma_0$:

- *tight budget ($\epsilon = 0.025$):* `RCB` achieves its best performance (approx. 0.35 error rate) across all prior variances. Notably, this matches the state-of-the-art performance of the Lasso Bandit baseline (Bastani & Bayati, 2020) (represented by the dotted line in Figure 2), despite the fact that `RCB` operates under much stricter conditions: it successfully enforces the DBIC constraints and does not require prior knowledge of non-zero feature counts.

- *intermediate budget ($\epsilon = 0.035$):* Performance degrades for weaker priors.

- *loose budget ($\epsilon = 0.045$):* The error rate exceeds 0.40 for all settings.

**Weighted Risk Score Analysis.** Table 2 breaks down the clinical decision quality by dosage stratum. The patient population exhibits significant class imbalance: 60% require Medium dosage, while 27% and 13% require Low and High dosages, respectively.

- *standard of care (Physician):* The fixed "Always Medium" strategy acts as a strong prior. It achieves 100% accuracy on the majority class (Medium) but fails completely on the critical Low and High tails. This yields a static baseline score of 0.20.

- *RCB performance:* By incentivizing exploration, `RCB` successfully identifies patients in the tails of the distribution. It attains correction rates of 50% for Low dosages and 5% for High dosages, while maintaining a robust 84% accuracy for the Medium group. Crucially, the rate of *extreme* errors (e.g., prescribing High to a Low patient) is limited to 2%, indicating the algorithm preserves safety constraints.

- *net clinical utility:* At a tight budget of $\epsilon = 0.025$, `RCB` achieves a weighted risk score of **0.291**, significantly outperforming the physician baseline (0.20). Even as the prior becomes weaker ($\epsilon = 0.035$), the score remains competitive (0.265). This confirms that `RCB` effectively trades off a marginal decrease in majority-class accuracy for significant gains in identifying high-risk minority patients.

**Scalability and Large $K$ Mitigation.** The cold-start sample complexity scales as $\mathcal{O}(K^3 \cdot d/\epsilon^2)$, which can be cost-

prohibitive for systems with many arms. To mitigate this in practice, we propose four strategies: (1) *Arm clustering*: grouping similar arms based on features to explore cluster representatives, reducing the effective $K$; (2) *Progressive exploration*: starting with a reduced arm set (e.g., top-$K$ by prior mean) and expanding as the system learns; (3) *Warm starting*: incorporating historical data or A/B tests as an informative prior to reduce the required cold-start samples; and (4) *Contextual arm elimination*: eliminating clearly suboptimal arms early to reduce the effective $K$ per context.

## 6. Conclusion

We introduced RCB, a framework that resolves the tension between myopic user incentives and long-term learning via a modular two-stage architecture. By replacing rigid posterior sampling with *Inverse Proportional Gap Sampling*, RCB allows for the integration of arbitrary offline regression oracles. A core contribution of this modularity is that non-linear oracles (e.g., neural networks) can be integrated seamlessly. While theoretical DBIC structures hold, verifying BIC with non-linear oracles may require bootstrap-based uncertainty quantification or conformal prediction, and regret bounds would scale with the complexity of the function class. Theoretically, we achieved a regret bound of $\tilde{\mathcal{O}}(\sqrt{KdT})$ and explicitly quantified the "Price of Incentivizing Exploration", deriving the cold-start cost required to satisfy the $\epsilon$-DBIC constraint. Empirical validation confirms that RCB significantly outperforms in identifying optimal treatments for high-risk groups. Future work will extend this modular approach to combinatorial semi-bandits, deep retrieval and ranking systems, and dynamically adjusting the budget $\epsilon$ to accommodate adaptive user behavior.

## Impact Statement

This work advances the field of contextual bandits by reconciling exploration with user incentives, potentially improving fairness in recommendation systems and personalized healthcare by identifying optimal outcomes for under-served populations. However, the reliance on information asymmetry to incentivize exploration raises ethical considerations regarding transparency and user autonomy. To mitigate risks in high-stakes domains, our framework enforces a rigorous DBIC constraint, ensuring recommendations remain within a safety margin of the user's myopic best interest. We emphasize that in clinical settings, such algorithms should function as decision-support tools for experts rather than autonomous agents. Practitioners must strictly validate prior beliefs and carefully calibrate the incentive budget to maintain these safety guarantees in deployment.

## Acknowledgement

We would like to thank the area chair and anonymous referees for their constructive suggestions that improve the paper. Xiaowu Dai acknowledges support from the National Science Foundation DMS 2515903, the National Institutes of Health R01DK142026, the National Institutes of Health dkNet AI Pilot Award (parent award U24DK097771), the Merck Biostatistics and Research Decision Sciences, and the Hellman Fellowship.

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

## A. Related Works

**Incentivized Exploration**. The study of incentivized exploration was initiated by (Kremer et al., 2014) and (Che & Horner, 2015), who focused on deriving Bayesian-optimal policies for two-action settings. While (Frazier et al., 2014) extended this to include monetary transfers, subsequent research has analyzed exploration incentives in diverse scenarios, including decentralized agents (Keller et al., 2005), dynamic pricing (Besbes & Zeevi, 2009), auctions (Ostrovsky & Schwarz, 2023; Babaioff et al., 2015), and human computation (Ho et al., 2014).

**Information Design** Another related work is Bayesian persuasion as introduced by (Kamenica & Gentzkow, 2011), focusing on a single round where the planner's signal is informed by the "history" of previous interactions. In exploring strategic information disclosure, (Rayo & Segal, 2010) investigated how planners can encourage better decision-making among agents by controlling information flows. The temporal aspect of information release is addressed by (Ely et al., 2015; Hörner & Skrzypacz, 2016), who studied the optimization of suspense and the commercial strategy of selling information over time, respectively. These contributions highlight different facets of information design.

**Bandit algorithms**. $\epsilon$-greedy (Auer et al., 2002; Chen et al., 2021; Han et al., 2022; Shi et al., 2022), explore-then-commit (Robbins, 1952; Abbasi-Yadkori et al., 2009; Li et al., 2022), upper confidence bound (UCB) (Lai & Robbins, 1985; Auer, 2002; Li et al., 2021; Wang et al., 2023), Thompson sampling (Thompson, 1933; Russo & Van Roy, 2014; Li et al., 2023), boostrap sampling (Kveton et al., 2019; Wang et al., 2020; Wu et al., 2022; Ramprasad et al., 2023), information directed sampling (Russo & Van Roy, 2014; Hao & Lattimore, 2022), inversely proportional to the gap sampling (Abe & Long, 1999; Foster & Rakhlin, 2020; Simchi-Levi & Xu, 2022), and betting (Waudby-Smith et al., 2022; Li et al., 2024).

**Applications in Medical Fields**. Patients' incentives are a significant barrier to conducting medical trials, especially large-scale ones for affordable treatments. BIC exploration represents a theoretical effort to overcome this challenge. Medical trials initially motivated the study of multi-arm bandits (MABs) and exploration-exploitation tradeoffs (Villar et al., 2015). However, disclosing information about the medical trial is necessary to meet the "informed consent" standards set by various regulations (Arango et al., 2016). In addition, medical trials, particularly those involving multiple treatments, underscore the relevance of BIC bandit exploration with multiple actions where traditional trials typically compare a new treatment against a placebo, but the designs incorporating multiple treatments are gaining practical importance and have been explored in biostatistics literature (Freidlin et al., 2008). BIC bandit exploration with contexts consideration is increasingly applied in adaptive trial designs, leveraging patients' "background information" to tailor treatments.

## B. Algorithm

---

**Algorithm 1** `Cold Start Stage`

---

**Input** $: K, \mathsf{N}, L, B, S, \{N_i(t)\}_{i \in [K]}, t = 1.$

1   STEP 1 - THE MOST POPULAR ARM SAMPLE COLLECTION (MPASC)

    **while** *there is no arm been pulled* $\mathsf{N}$ *times* **do**

2        Agent $p_t$ is recommended with arm $i = \operatorname{argmax}_{j \in [K]} \mathbb{E}[\mu(x_t, j)]$ and receives reward $y_{t,i}$.

         The platform updates pulls and rewards: $N_i(t) \leftarrow N_i(t-1) + 1$, $S_i \leftarrow S_i \cup (x_t, y_{t,i})$.

         If $N_i(t) = \mathsf{N}$, add $i$ to $B_t$. $t \leftarrow t + 1$. STEP 1 stopped.

         Update $t \leftarrow t + 1$.

3   STEP 2 - REST ARM SAMPLE COLLECTION (RASC)

    **while** *there exists an arm* $i$ *such that the number of pulled* $N_i(t)$ *has not reached* $\mathsf{N}$ **do**

4        Samples $q_t \sim \text{Ber}(1/L)$.

         **if** $q_t = 1$ **then**

5            $p_t$ is recommended to explore with the arm $\widetilde{a}_t$ based on Eq.4 and receives $y_{t,\widetilde{a}_t}$.

             Updates $N_{\widetilde{a}_t}(t) \leftarrow N_{\widetilde{a}_t}(t-1) + 1$ and dataset $S_{\widetilde{a}_t} \leftarrow S_{\widetilde{a}_t} \cup (x_t, y_{t,\widetilde{a}_t})$.

             If $N_{\widetilde{a}_t}(t) = \mathsf{N}$, add $\widetilde{a}_t$ to $B_t$.

6          **else**

7            $p_t$ is recommended to exploit with the arm $a_t^*$ based on Eq.5 and receives $y_{t,a_t^*}$.

8        Update $t \leftarrow t + 1$.

---

---

**Algorithm 2** `Exploitation stage`

---

**Input** : $S$, epochs $m_0, m_1$, function class $\mathcal{F}$, learning algorithm $\text{Off}_\mathcal{F}$, confidence level $\delta$.

9 **for** *epoch* $m \in [m_0, m_1]$ **do**

10      Set $\gamma_m = 4\sqrt{K/\mathcal{E}_{\mathcal{F},\delta}(|\mathcal{T}_{m-1}|)}$.

        Feed $m-1$ epoch's data $W_{\mathcal{T}_{m-1}}$ into the $\text{OffPos}$ and get $\{\widehat{\beta}_{m,i}\}_{i\in[K]}$.

        **for** $t \in \mathcal{T}_m$ **do**

11            Agent $p_t$ arrives with covariate $x_t$. Compute estimate $\widehat{\mu}_{m(t)}(x_t, i) = x_t^\top \widehat{\beta}_{m,i}, \forall i \in [K]$.

           Obtain the optimal arm $b_t = \text{argmax}_{i\in[K]} \widehat{\mu}_{m(t)}(x_t, i)$.

           Sample $a_t \sim p_m(i)$ according to Eq.6 and observe reward $y_t(a_t)$.

---

# C. DBIC Property

## C.1. Proof of Theorem 1 - Cold Start Stage

*Proof.* To guarantee the DBIC property for the cold start of RCB, it suffices to have a lower bound on parameter $L$ to avoid too many samples wasted in the cold start stage.

The cold start stage can be split into $K$ phases and each phase last $L$N round in expectation based on the algorithm design except the most popular arm. Although the first phase (most popular arm) last unknown rounds, it usually lasts a pretty short period. So in the following analysis, we ignore the DBICproperty in the initial sample collection stage (MPASC stage).

Due to the design of cold start stage, agents are unaware which phase they belong to, they are only aware they have $1/L$ probability to be chosen in the cold start stage. We first argue that for each agent $p_t$ in phase $l \in [2, K]$ (except the MPASC), she has no incentive not to follow the recommended arm.

**(1).** If agent $p_t$ is recommended with the arm $j \neq \widetilde{a}_t$, then she knows since this arm $j$ is the organic arm $a_t^*$ and is not the promoted arm; so by the definition of the organic arm, it is DBIC for the agent to follow it.

**(2).** If agent $p_t$ is recommended with the arm $\widetilde{a}_t$ and does not want to deviate to some other arms $j \neq \widetilde{a}_t$. That is to say, we need to prove that when the platform recommends arm $i$, the agent $p_t$ has no incentive to deviate the current recommendation arm $i$ to other arm $j$ in expected reward. From the user's perspective, the platform needs to demonstrate this,

$$\mathbb{E}[\mu(x_t, i) - \mu(x_t, j)|I_t = i]\Pr(I_t = i) \geq 0. \tag{C.1}$$

Denote the time dependent posterior gap $G_{tij} := \mathbb{E}[\mu(x_t, i) - \mu(x_t, j)|S_{B_t}]$ where arm $i$ is the recommended arm by RCB and $j \neq i$, and the corresponding minimal posterior gap $G_t(i) = \min_{j\neq i} G_{tij}$. The $G_{tij}$ represents the posterior gap between arm $i$ and arm $j$ at time $t$. The $G_t(i)$ represents the minimal gap given the current accumulative samples which is composed of two cases: (1) $G_t(i) > 0$, that means arm $i$ is the posterior best arm. (2) $G_t(i) \leq 0$, that means arm $i$ is not the posterior best arm.

To satisfy the $\epsilon$-DBICproperty, we need the Eq.C.1 satisfied. By the law of iterated expectations $E[X] = E[E[X|Y]]$, we have

$$\begin{aligned}
&\mathbb{E}[\mu(x_t, i) - \mu(x_t, j)|I_t = i]\Pr(I_t = i) \\
&= \mathbb{E}[\mathbb{E}[\mu(x_t, i) - \mu(x_t, j)|S_{B_t}]|I_t = i]\Pr(I_t = i) \\
&= \mathbb{E}[G_{tij}|I_t = i]\Pr(I_t = i) > -\epsilon.
\end{aligned} \tag{C.2}$$

Define two events $Q_{t,1} = \{q_t = 1\}$ and $Q_{t,0} = \{q_t = 0\}$, representing agent $p_t$ is recommended with the promoted arm or organic arm respectively. Thus, there are two disjoint events under which agent $p_t$ is recommended arm $i$, either $E_{t1} = \{G_t(i) > 0\}$ or $E_{t2} = \{G_t(i) \leq 0\} = \{G_t(i) \leq 0 \text{ and } p_t \in Q_{t,1}\}$. For notation simplicity, we denote $E_1 = E_{t1}$ and $E_2 = E_{t2}$. The reason $\{G_t(i) \leq 0\} = \{G_t(i) \leq 0 \text{ and } p_t \in Q_{t,1}\}$ is because $G_t(i) \leq 0$ happens only when $p_t \in Q_{t,1}$. So the above equation is equivalent to prove

$$\mathbb{E}[G_{tij}|I_{p_t} = i]\Pr(I_t = i) = \mathbb{E}[G_{tij}|E_1]\Pr(E_1) + \mathbb{E}[G_{tij}|E_2]\Pr(E_2) > 0. \tag{C.3}$$

We observe that $\Pr(E_2) = \Pr(p_t \in Q_{t,1}|G_t(i) \leq 0)\Pr(G_t(i) \leq 0) = \Pr(G_t(i) \leq 0)/L_t$, where $q_t \sim \text{Ber}(1/L_t)$ and is time dependent and independent of other random variables. Since the event $p_t \in Q$ is independent of $G_{tij}$ and agent $p_t$ in

$Q_{t,1}$ is randomly selected according to the Bernoulli distribution with expectation $1/L_t$. Therefore, we get:

$$
\begin{aligned}
\mathbb{E}[\mu(x_t, i) &- \mu(x_t, j)|I_t = i]\mathrm{Pr}(I_t = i)\\
&= \mathbb{E}[G_{tij}|E_1]\mathrm{Pr}(E_1) + \mathbb{E}[G_{tij}|E_2]\mathrm{Pr}(E_2)\\
&= \mathbb{E}[G_{tij}|G_t(i) > 0]\mathrm{Pr}(G_t(i) > 0) + \mathbb{E}[G_{tij}|G_t(i) \le 0 \text{ and } p_t \in Q_{t,1}]\frac{1}{L_t}\mathrm{Pr}(G_t(i) \le 0)\\
&= \mathbb{E}[G_{tij}|G_t(i) > 0]\mathrm{Pr}(G_t(i) > 0) + \frac{1}{L_t}\mathbb{E}[G_{tij}|G_t(i) \le 0]\mathrm{Pr}(G_t(i) \le 0),
\end{aligned}
\tag{C.4}
$$

where the second equation holds by the independent property. By the fact that $\mathbb{E}[G_{tij}] = \mathbb{E}[G_{tij}|G_t(i) \le 0]\mathrm{Pr}(G_t(i) \le 0) + \mathbb{E}[G_{tij}|G_t(i) > 0]\mathrm{Pr}(G_t(i) > 0)$, so the above equation becomes

$$
\begin{aligned}
&= \mathbb{E}[G_{tij}|G_t(i) > 0]\mathrm{Pr}(G_t(i) > 0) + \frac{1}{L_t}\left(\mathbb{E}[G_{tij}] - \mathbb{E}[G_{tij}|G_t(i) > 0]\mathrm{Pr}(G_t(i) > 0)\right)\\
&= (1 - \frac{1}{L_t})\mathbb{E}[G_{tij}|G_t(i) > 0]\mathrm{Pr}(G_t(i) > 0) + \frac{1}{L_t}\mathbb{E}[G_{tij}].
\end{aligned}
\tag{C.5}
$$

We know $\mathbb{E}[G_{tij}] = \mathbb{E}[\mathbb{E}[\mu(x_t, i) - \mu(x_t, j)|S_{B_t}]] = \mathbb{E}[\mu(x_t, i) - \mu(x_t, j)] = x_t^\mathsf{T}\beta_{i,0} - x_t^\mathsf{T}\beta_{j,0} = \mu_0(t, i) - \mu_0(t, j)$. Thus, the above equation will be

$$
= (1 - \frac{1}{L_t})\mathbb{E}[G_{tij}|G_t(i) > 0]\mathrm{Pr}(G_t(i) > 0) + \frac{1}{L_t}(\mu_0(t, i) - \mu_0(t, j)).
\tag{C.6}
$$

To make the process be $\epsilon$-DBIC, we need $\mathbb{E}[\mu(x_t, i) - \mu(x_t, j)|I_t = i]\mathrm{Pr}(I_t = i) > -\epsilon$. Since we know $G_{tij} > G_t(i)$ by definition, so we have $\mathbb{E}[G_{tij}|G_t(i) > 0] > \mathbb{E}[G_t(i)|G_t(i) > 0]$. To combine them all, we get

$$
\ge (1 - \frac{1}{L_t})\mathbb{E}[G_t(i)|G_t(i) > 0]\mathrm{Pr}(G_t(i) > 0) + \frac{1}{L_t}(\mu_0(t, i) - \mu_0(t, j)) \ge -\epsilon.
\tag{C.7}
$$

Thus, $\forall i, j \in [K]$, it suffices to pick $L_t$ at time $t$ such that:

$$
\begin{aligned}
L_t &\ge 1 - \frac{\mu_0(t, i) - \mu_0(t, j) + \epsilon}{\mathbb{E}[G_t(i)|G_t(i) > 0]\mathrm{Pr}(G_t(i) > 0) + \epsilon}\\
&= 1 + \frac{\mu_0(t, j) - \mu_0(t, i) - \epsilon}{\mathbb{E}[G_t(i)|G_t(i) > 0]\mathrm{Pr}(G_t(i) > 0) + \epsilon},
\end{aligned}
\tag{C.8}
$$

Thus we need,

$$
L_t \ge 1 + \frac{\overline{\Delta}_t^0 - \epsilon}{\tau_{\mathcal{P}_{0,t}}\rho_{\mathcal{P}_{0,t}} + \epsilon},
\tag{C.9}
$$

where $\overline{\Delta}_t^0 = \max_{i \ne j}[\mu_0(t, j) - \mu_0(t, i)]$, and $\mathbb{E}[G_t(i)|G_t(i) > 0]\mathrm{Pr}(G_t(i) > 0) \ge \tau_{\mathcal{P}_{0,t}}\rho_{\mathcal{P}_{0,t}}$.

By the design of the cold start stage, we know that arm $i$ is the platform recommended arm and arm $j$ is the arm agent $p_t$ potentially wants to deviate to. Therefore, based on the prior knwoledge, $\mu_0(t, j) \ge \mu_0(t, i)$. Since this $L_t$ is time dependent, to get a time uniform $L$ to let all agents have the DBIC property, we need

$$
\max_t L_t = 1 + \frac{\overline{\Delta}^0 - \epsilon}{\tau_{\mathcal{P}_0}\rho_{\mathcal{P}_0} + \epsilon},
\tag{C.10}
$$

where $\overline{\Delta}^0 = \max_t \overline{\Delta}_t^0$ and we know $\overline{\Delta}^0 \le 1$, and $\tau_{\mathcal{P}_0} = \min_t \tau_{\mathcal{P}_{0,t}}, \rho_{\mathcal{P}_0} = \min_t \rho_{\mathcal{P}_{0,t}}$. So we have $L$ needs to be at least

$$
L \ge 1 + \frac{1 - \epsilon}{\tau_{\mathcal{P}_0}\rho_{\mathcal{P}_0} + \epsilon}.
\tag{C.11}
$$

By selecting the time uniform $L$, we have the DBIC property.

$\square$

## C.2. Proof of Theorem 1 - Exploitation stage

*Proof.* To satisfy the DBICproperty, which is any agent $p_t$ who is recommended arm $i$ ($I_t = i$) does not to want to switch to some other arm $j$ in expectation. Besides, we assert that when the platform satisfies the DBICproperty at the cold start stage and the DBICproperty also holds when we have a minimum requirement of N, then in the following epochs, the RCB algorithm will automatically satisfy the DBICin the Exploitation stage. More formally, we need that

$$\mathbb{E}[\mu(x_t, i) - \mu(x_t, j)|I_t = i]\Pr(I_t = i) \geq -\epsilon/K, \forall t \in \text{Exploitation stage.} \tag{C.12}$$

Similarly to the construction of $L$ in the previous analysis, we denote the time dependent posterior gap $G_{tij} := \mathbb{E}[\mu(x_t, i) - \mu(x_t, j)|S_*]$ where arm $i$ is the recommended arm by RCB and $j \neq i$, where $S_*$ is the dataset collected in the the cold start stage. The corresponding minimal posterior gap $G_t(i) = \min_{j \neq i} G_{tij}$. The $G_{tij}$ represents the posterior gap between arm $i$ and arm $j$ at time $t$. The $G_t(i)$ represents the minimal gap given the current accumulative samples which is composed of two cases: (1) If $G_t(i) > 0$, that means arm $i$ is the best arm in terms of the posterior. (2) If $G_t(i) \leq 0$, that means arm $i$ is not the posterior best arm. Recall the definition of $G_t(i)$, it suffices to show that

$$\begin{aligned}
\mathbb{E}[G_t(i)|I_t = i] &= \mathbb{E}\left[\mathbb{E}[\mu(x_t, i) - \max_{j \in [K]/i} \mu(x_t, j)|S_*]|I_t = i\right] \\
&= \mathbb{E}\left[\mathbb{E}[\mu(x_t, i)|S_*] - \max_{j \in [K]/i} \mathbb{E}[\mu(x_t, j)|S_*]|I_t = i\right].
\end{aligned} \tag{C.13}$$

Let $S_*$ be the data set collected by the algorithm by the beginning of Exploitation stage. The reward gap can be decomposed as

$$\begin{aligned}
&\mathbb{E}\left[\mathbb{E}[\mu(x_t, i)|S_*] - \max_{j \in [K]/i} \mathbb{E}[\mu(x_t, j)|S_*]|I_t = i\right]\Pr(I_t = i) \\
&= \underbrace{\Pr(i = b_t)\mathbb{E}\left[\mathbb{E}[\mu(x_t, i)|S_*] - \max_{j \in [K]/i} \mathbb{E}[\mu(x_t, j)|S_*]|i = b_t\right]}_{\text{Part I Reward Gap}} \\
&\quad + \underbrace{\Pr(i \neq b_t)\mathbb{E}\left[\mathbb{E}[\mu(x_t, i)|S_*] - \max_{j \in [K]/i} \mathbb{E}[\mu(x_t, j)|S_*]|i \neq b_t\right]}_{\text{Part II Reward Gap}},
\end{aligned} \tag{C.14}$$

where $b_t$ is the highest posterior mean arm $b_t = \text{argmax}_{j \in [K]} \mathbb{E}[\mu(x_t, j)|S_*]$.

**Part I Reward Gap:** The platform selects the highest posterior mean reward arm $b_t = \text{argmax}_{j \in [K]} \mathbb{E}[\mu(x_t, j)|S_*] = \text{argmax}_{j \in [K]} \widehat{\mu}_m(x_t, j)$ according to the Algorithm 2's design with probability $\Pr(I_t = b_t) = 1 - \sum_{i \neq b_t} \frac{1}{K + \gamma_m u_i}$, where $u_i = \widehat{\mu}_m(x_t, b_t) - \widehat{\mu}_m(x_t, i)$. Denote $G_t(b_t)$ as the minimal optimal posterior gap $\mathbb{E}[\mu(x_t, b_t)|S_*] - \max_{j \in [K]/b_t} \mathbb{E}[\mu(x_t, j)|S_*]$, which is the gap between the highest posterior mean utility and second highest posterior mean utility. By the sampling design of RCB and $\gamma_m > 0, \forall m \geq m_0$, we get that $p(b_t) \geq 1/K$, where $p(b_t)$ is the probability of selecting the highest posterior mean arm.

$$\text{Part I Reward Gap} \geq \frac{1}{K} G_t(b_t). \tag{C.15}$$

**Part II Reward Gap:** According to the sampling structure, it has the probability that the platform recommended arm is not $b_t$, we have

$$\begin{aligned}
\text{Part II Reward Gap} &= \Pr(i \neq b_t)\mathbb{E}\left[\mathbb{E}[\mu(x_t, i)|S_*] - \max_{j \neq i} \mathbb{E}[\mu(x_t, j)|S_*]|i \neq b_t\right] \\
&= \sum_{i \neq b_t} p_t(i)\left[\mathbb{E}[\mu(x_t, i)|S_*] - \mathbb{E}[\mu(x_t, b_t)|S_*]\right] \\
&= -\sum_{i \neq b_t} p_t(i)\left[\mathbb{E}[\mu(x_t, b_t)|S_*] - \mathbb{E}[\mu(x_t, i)|S_*]\right] \\
&= -r_t
\end{aligned} \tag{C.16}$$

where $r_t = \sum_{i \neq b_t} p_t(i)(\mathbb{E}[\mu(x_t, b_t)|S_*] - \mathbb{E}[\mu(x_t, i)|S_*])$. Therefore, to achieve DBICproperty, we can lower bound the following term,

$$\mathbb{E}[\mu(x_t, i) - \mu(x_t, j)|I_t = i]\Pr(I_t = i) \geq \frac{G_t(b_t)}{K} - r_t \geq \frac{G_t(b_t)}{K} - \frac{K}{\gamma_m} \qquad \text{(C.17)}$$

The $G_t(b_t)/K$ is each step's expected gain and $r_t$ is each step's expected loss, and by Lemma 7, we have $r_t \leq K/\gamma_m$ used in the last inequality. In order to satisfy the DBIC property, we need

$$\frac{G_t(b_t)}{K} - \frac{K}{\gamma_m} > -\frac{\epsilon}{K} \qquad \text{(C.18)}$$

which is equivalent to need

$$\gamma_m(\epsilon) \geq \frac{K^2}{G_t(b_t) + \epsilon}. \qquad \text{(C.19)}$$

That is, in order to satisfy the DBIC property, we need the spread parameter at each epoch $m(\geq m_0)$ is at least greater than $\gamma_m(\epsilon)$. Here $\tau_m = 2^m$ is the time step where epoch $m$ stops. $\mathcal{E}_{\mathcal{F},\delta}(m-1)$ represents the prediction error in the functional class $\mathcal{F}$ when using training data collected in epoch $m-1$ that is in the time interval $(\tau_{m-2}, \tau_{m-1}]$. Based on the offline learning's result from Definition 2 given the epoch $m$, we have $\gamma_{m_0} = c\sqrt{K/\mathcal{E}_{\mathcal{F},\delta}(\tau_{m_0-1} - \tau_{m_0-2})}$. So we can derive the requirement of the minimum prediction error at epoch $m_0$. We need

$$\gamma_{m_0} \geq \gamma_m(\epsilon)$$
$$c\sqrt{\frac{K}{\mathcal{E}_{\mathcal{F},\frac{\delta}{2K^2}}(\tau_{m_0-1} - \tau_{m_0-2})}} \geq \frac{K^2}{G_t(b_t) + \epsilon}$$
$$\mathcal{E}_{\mathcal{F},\frac{\delta}{2K^2}}(\tau_{m_0-1} - \tau_{m_0-2}) \leq \frac{c^2(G_t(b_t) + \epsilon)^2}{K^3} \qquad \text{(C.20)}$$
$$\frac{c_3\sigma^2 d}{\phi_0 n} \leq \frac{c^2(G_t(b_t) + \epsilon)^2}{K^3}$$
$$n \geq \frac{(\sigma^2 d + 1)K^3}{\phi_0(G_t(b_t) + \epsilon)^2}$$

where $\mathcal{E}_{\mathcal{F},\frac{\delta}{2K^2}}(\tau_{m_0-1} - \tau_{m_0-2})$ is the prediction error with training sample size with $n = \tau_{m-1} - \tau_{m-2}$, which bounds the squared $L_2$ distance between $\hat{\mu}$ and $\mu$ on the test data sampled following the same data generation process as the training data. For the forth inequality, based on Corollary 1, we need the minimum sample size as $\mathsf{N}(\epsilon) = \frac{(\sigma^2 d+1)K^3}{\phi_0(G_t(b_t)+\epsilon)^2}$. We have $\tau_m = 2^m$, $\tau_{m-1} - \tau_{m-2} = 2^{m-1} - 2^{m-2} = 2^{m-2}$. By the minimum sample size requirement for the cold start stage's $\mathsf{N}(\epsilon)$ for each arm, we know in Exploitation stage, the starting epoch $m_0$ should be

$$\mathsf{N} \leq \tau_{m-1} - \tau_{m-2},$$
$$\log_2 \mathsf{N} \leq m - 2, \qquad \text{(C.21)}$$
$$m \geq m_0 = \lceil 2 + \log_2 \frac{(\sigma^2 d + 1)K^3}{\phi_0(\tau_{\mathcal{P}_*} + \epsilon)^2} \rceil.$$

where $\tau_{\mathcal{P}_*}$ is the minimum posterior mean gap based on Assumption 1. $\qquad \square$

## D. Prediction Error of Ridge Regression with Random Design

From (Mourtada & Rosasco, 2022), we have the following lemmas of the prediction error of ridge regression with random design.

**Lemma 1.** *Assume the noise has gaussian distribution, then the excess risk bound is*

$$\mathbb{E}\left[\left\|\hat{\beta} - \beta\right\|_{\Sigma}^2\right] \leq \left(1 + \frac{R^2}{\lambda n}\right)^2 \inf_{\beta \in \mathbb{R}^d}\{L(\beta) + \lambda\|\beta\|^2 - L(\beta^*)\} + \left(1 + \frac{R^2}{\lambda n}\right)\frac{\sigma^2 Tr[(\Sigma + \lambda)^{-1}\Sigma]}{n} \qquad \text{(D.1)}$$

*where $\|X\|_2 \leq R$ and risk $L(\beta) = \mathbb{E}[(Y - \langle\beta, X\rangle)^2]$.*

**Lemma 2.** *For every $\lambda > 0$, we have*

$$\inf_{\beta \in \mathbb{R}^d} \{L(\beta) + \lambda \|\beta\|^2 - L(\beta^*)\} = \lambda \left\| (\Sigma + \lambda)^{-1/2} \Sigma^{1/2} \beta^* \right\|^2 \leq \lambda \|\beta^*\|^2 . \tag{D.2}$$

**Corollary 1.** *The prediction error can be upper bounded bounded by*

$$\mathbb{E}\left[ (\widehat{\beta}^{\mathsf{T}} X_t - \beta^{\mathsf{T}} X_t)^2 \right] \leq \mathbb{E}\left[ \left\| \widehat{\beta} - \beta \right\|_{\Sigma}^2 \right] \mathbb{E}\left[ \|X_t\|_{\Sigma^{-1}}^2 \right] \leq \frac{R^2}{\lambda_{\min}(\Sigma)} \mathbb{E}\left[ \left\| \widehat{\beta} - \beta \right\|_{\Sigma}^2 \right] \leq \frac{c_3 \sigma^2 d}{\phi_0 n} .$$

*Proof.* By Lemma 1 and Lemma 2, we have

$$\begin{aligned}
\mathbb{E}\left[ \left\| \widehat{\beta} - \beta \right\|_{\Sigma}^2 \right] &\leq \left( 1 + \frac{R^2}{\lambda n} \right)^2 \lambda \|\beta^*\|^2 + \left( 1 + \frac{R^2}{\lambda n} \right) \frac{\sigma^2 d}{n} \\
&\leq \left( 1 + \frac{1}{c_1} \right)^2 \frac{c_1}{n} + \left( 1 + \frac{1}{c_1} \right) \frac{\sigma^2 d}{n}
\end{aligned} \tag{D.3}$$

Assume that $\|\beta\|_2 \leq 1$, $R \leq 1$ and $\lambda = \frac{c_1}{n}$. So when $n \geq N = \frac{1}{c_1^2 (c_2 - 1)^2}$ and denote $c_2 = (1 + \frac{1}{c_1})^2$. So when $n \geq N$, we have

$$\begin{aligned}
\mathbb{E}\left[ \left\| \widehat{\beta} - \beta \right\|_{\Sigma}^2 \right] &\leq \frac{c_1 c_2}{n} + \frac{c_2 \sigma^2 d}{n} \\
&\leq \frac{c_1 c_2 + c_2 \sigma^2 d}{n} \\
&\leq c_3 \frac{\sigma^2 d}{n}
\end{aligned} \tag{D.4}$$

where we define $c_3 \sigma^2 d = c_1 c_2 + c_2 \sigma^2 d$ for $c_3 > 0$. Since we know $\lambda_{\min}(\Sigma) \geq \phi_0$, so the prediction error can be upper bounded by

$$\mathbb{E}\left[ (\widehat{\beta}^{\mathsf{T}} X_t - \beta^{\mathsf{T}} X_t)^2 \right] \leq \frac{c_3 \sigma^2 d}{\phi_0 n} \tag{D.5}$$

$\square$

# E. Proof of No Regret Learning

We first denote $\Psi := \mathcal{A}^{\mathcal{X}}$ as the universal policy space, which contains all possible policies. Here we assume that $|\mathcal{X}| < \infty$ but allows $|\mathcal{X}|$ to be arbitrarily large. Focusing on such a setting enables us to highlight important ideas and key insights without the need to invoke measure theoretic arguments, which are necessary for infinite/uncountable $\mathcal{X}$. At epoch $m(t)$, $m = m(t)$ if $t$ is clear, and $p_t(\cdot) = p_m(\cdot|x_t)$. We next analyze the following virtual process at round $t$ in epoch $m(t)$. Here we use a novel virtual probability distribution $Q_m(\cdot)$ to analyze the $p_t(\cdot)$'s effect over the regret. There are three steps:

1. Algorithm samples $\pi_t \sim Q_m(\cdot)$, where $\pi_t : \mathcal{X} \to \mathcal{A}$ is a deterministic policy, and $Q_m(\cdot) : \mathcal{A}^{\mathcal{X}} \to$ Probability Measure (a probability distribution over all policies in $\mathcal{A}^{\mathcal{X}}$).

2. At time $t$, $x_t \sim \mathcal{P}_X$.

3. Algorithm selects $a_t = \pi_t(x_t)$.

Note that at round $t$, $Q_m(\cdot)$ is a stationary distribution which has already been determined at the beginning of epoch $m$. How to construct this $Q_m(\cdot)$? For any policy $p_m(\cdot|\cdot)$, we can construct a unique product probability measure $Q_m(\cdot)$ on $\Psi$ such that $Q_m(\pi) = \prod_{x \in \mathcal{P}_X} p_m(\pi(x)|x)$ for all $\pi \in \Psi$. This product measure $Q_m(\cdot)$ ensures that for every

$$p_m(a|x) = \sum_{\pi \in \Psi} \mathbb{I}\{\pi(x) = a\} Q_m(\pi(x)). \tag{E.1}$$

That is, for any arbitrary context $x \in \mathcal{X}$, the algorithm's recommended action generated by $p_m(\cdot|x)$ is probabilistically equivalent to the action generated by $Q_m(\cdot)$ through this virtual process. Since $Q_m(\cdot)$ is a dense distribution over all deterministic polices in the universal policy space, we refer to $Q_m(\cdot)$ as the "equivalent randomized policy" induced by

$p_m(\cdot|\cdot)$. Since $p_m(\cdot|\cdot)$ is completed determined by $\gamma_m$ and $\widehat{\mu}_m$, we know that $Q_m(\cdot)$ is also completely determined by $\gamma_m$ and $\widehat{\mu}_m$. We emphasize that the Exploitation stage does not actually compute $Q_m(\cdot)$, but implicit maintains $Q_m(\cdot)$ through spread parameter $\gamma_m$ and estimated posterior mean $\widehat{\mu}_m$, so called virtual process. That is important, as even when $\mathcal{X}$ is known to the learner, computing the product measure $Q_m(\cdot)$ requires $\Omega(|\mathcal{X}|)$ computational cost which is intractable for large $\mathcal{X}$.

To get the regret upper bound, we need following notations. For any action selection kernel $p$ and any policy $\pi$, let's define the following terms:

1. **Reward** $R_t(\pi)$: defines the expected reward in the measure of $\mu$ if it follows the policy $\pi$ to select the action $\pi(x_t)$ with respect to distribution $\mathcal{P}_X$: $R_t(\pi) = \mathbb{E}_{x_t \sim \mathcal{D}_{\mathcal{X}}}[\mu(x_t, \pi(x_t))]$

2. **Reward** $\widehat{R}_t$: defines the expected reward in the measure of empirical $\widehat{\mu}_{m(t)}$ if follows the policy $\pi$ to select the action $\pi(x_t)$ with respect to distribution $\mathcal{P}_X$: $\widehat{R}_t(\pi) = \mathbb{E}_{x_t \sim \mathcal{D}_{\mathcal{X}}}[\widehat{\mu}_{m(t)}(x_t, \pi(x_t))]$.

3. **Regret** $\mathrm{Reg}(\pi)$: defines the expected regret in the measure of $\mu$ if it follows the policy $\pi$ to select the action $\pi(x_t)$ with respect to distribution $\mathcal{P}_X$: $\mathrm{Reg}(\pi) = R_t(\pi_\mu) - R_t(\pi)$.

4. **Regret** $\widehat{\mathrm{Reg}}_t(\pi)$: defines the expected regret in the measure of empirical $\widehat{\mu}_{m(t)}$ if it follow the policy $\pi$ to select the action $\pi(x_t)$ with respect to distribution $\mathcal{P}_X$: $\widehat{\mathrm{Reg}}_t(\pi) = \widehat{R}_t(\pi_{\widehat{\mu}_{m(t)}}) - \widehat{R}_t(\pi)$.

where $\pi_{\widehat{\mu}_{m(t)}}$ is the policy selects the action $b_t = \mathrm{argmax}_{i \in [K]} \widehat{\mu}_{m(t)}(x_t, i)$ according to Eq.6.

Besides, for any probability kernel $p_m$ and any policy $\pi(\cdot)$, let $V(p_m, \pi)$ denote the expected inverse probability

$$V(p_m, \pi) = \mathbb{E}_{x_t \sim \mathcal{D}_{\mathcal{X}}}\left[\frac{1}{p_m(\pi(x_t)|x_t)}\right] \tag{E.2}$$

and define $\mathcal{V}_t(\pi)$ as the maximum expected inverse probability over the Exploitation stage,

$$\mathcal{V}_t(\pi) = \max_{m_0 \leq m \leq m(t)-1} V(p_m, \pi) \tag{E.3}$$

### E.1. Key Lemmas

**Lemma 3** (Azuma-Hoeffding Inequality). *Let $\{D_k, \mathcal{F}_k\}_{k=1}^\infty$ be a martingale difference sequence for which there are constants $\{(a_{k,b_k})\}_{k=1}^n$, such that $D_k \in [a_{k,b_k}]$ almost surely for all $k = 1, 2, ..., n$. Then, for all $t \geq 0$, $\mathrm{Pr}[|\sum_{k=1}^n D_k| \geq t] \leq 2\exp[-\frac{2t^2}{\sum_{k=1}^n (b_k - a_k)^2}]$.*

**Lemma 4.** *$\forall t \in [\tau_{m-1} + 1, \tau_m]$, with probability at least $1 - \delta/2m^2$, we have*

$$\mathbb{E}_{x_t, a_t}\left[\left(\widehat{\mu}_{m(t)}(x_t, a_t) - \mu(x_t, a_t)\right)^2 | \mathfrak{S}_{t-1}\right] \leq \mathcal{E}_{\mathcal{F}, \delta/(2m^2)}(\tau_{m-1} - \tau_{m-2}) = \frac{16K}{\gamma_m^2} \tag{E.4}$$

*where $\tau_m = 2^m$. Therefore, the following event $\Lambda_2$ holds with probability at least $1 - \delta/2$:*

$$\Lambda_2 := \left\{\forall t \geq \tau_{m_0}, \mathbb{E}_{x_t, a_t}\left[\left(\widehat{\mu}_{m(t)}(x_t, a_t) - \mu(x_t, a_t)\right)^2 | \mathfrak{S}_{t-1}\right] \leq \frac{16K}{\gamma_m^2}\right\}. \tag{E.5}$$

*Proof.* Note that Algorithm 2 always collects $(x_t, a_t; y_t(a_t))$-type data used for OffPos algorithm to conduct offline training, where $(x_t, y_t) \sim \mathcal{D}$ and $a_t \sim p_{m(t)-1}(\cdot|x_t)$ based on epoch $m(t) - 1$ collected data. Based on the prediction error of the OffPos algorithm provided in 2, we have $\forall t \in [\tau_{m-1} + 1, \tau_m]$,

$$\mathbb{E}_{x_t, a_t}\left[\left(\widehat{\mu}_{m(t)}(x_t, a_t) - \mu(x_t, a_t)\right)^2 | \mathfrak{S}_{t-1}\right] = \mathbb{E}_{x_t \sim \mathcal{P}_X, a_t \sim p_{m(t)-1}(\cdot|x_t)}\left[\left(\widehat{\mu}_{m(t)}(x_t, a_t) - \mu(x_t, a_t)\right)^2 | p_{m(t)-1}\right]$$

$$\leq \mathcal{E}_{\mathcal{F}, \delta/(2m^2)}(\tau_{m-1} + 1 - \tau_{m-2} - 1) = \frac{16K}{\gamma_m^2}, \tag{E.6}$$

where last the inequality simply follows from Lemma 4.1 and Lemma 4.2 from (Agarwal et al., 2012). $\square$

As we mentioned in previous, a starting point of our proof of regret upper bound is to translate the action selection kernel $p_m(\cdot|\cdot)$ into an equivalent distribution over policies $Q_m(\cdot)$. The following lemma provides a justification of such translation by showing the existence of an equivalent $Q_m(\cdot)$ for every $p_m(\cdot|\cdot)$. Here we refer Lemma 3 from (Simchi-Levi & Xu, 2022) in the following Lemma.

**Lemma 5.** *Fix any epoch $m \geq m_0$. The action selection scheme $p_m(\cdot|\cdot)$ is a valid probability kernel $\mathcal{B}(\mathcal{A}) \times \mathcal{X} \to [0,1]$ over epoch $m$. There exists a probability measure $Q_m$ on $\Psi$ such that*

$$\forall a_t \in \mathcal{A}, \forall x_t \in \mathcal{X}, p_m(a_t|x_t) = \sum_{\pi \in \Psi} \mathbb{I}\{\pi(x_t) = a_t\} Q_m(\pi) \tag{E.7}$$

The following Lemma demonstrates $y_t(\pi_\mu) - y_t(a_t) - \sum_{\pi \in \Psi} Q_m(\pi) \text{Reg}(\pi)$ is a martingale difference sequence with respect to $\mathfrak{S}_t$.

**Lemma 6.** *Fix any epoch $m \geq m_0 \in \mathbb{N}$, for any round $t$ in epoch $m$, we have:*

$$\mathbb{E}_{x_t, y_t, a_t}\left[y_t(\pi_\mu) - y_t(a_t)|\mathfrak{S}_{t-1}\right] = \sum_{\pi \in \Psi} Q_m(\pi) Reg(\pi) \tag{E.8}$$

*Proof.* By the definition of $\mathbb{E}[y_t(a_t)]$, we have

$$
\begin{aligned}
&\mathbb{E}_{x_t, y_t, a_t}\left[y_t(\pi_\mu(x_t)) - y_t(a_t)|\mathfrak{S}_{t-1}\right] \\
&= \mathbb{E}_{x_t, a_t}\left[\mu(x_t, \pi_\mu(x_t)) - \mu(x_t, a_t)|\mathfrak{S}_{t-1}\right] \\
&= \mathbb{E}_{x_t \sim \mathcal{P}_X, a_t \sim p_{m(t)}(\cdot|x)}\left[\mu(x_t, \pi_\mu(x_t)) - \mu(x_t, a_t)\right] \\
&= \mathbb{E}_{x_t \sim \mathcal{D}_\mathcal{X}}\left[\sum_{a_t \in \mathcal{A}} p_{m(t)}(a_t|x_t)\Big(\mu(x_t, \pi_\mu(x_t)) - \mu(x_t, a_t)\Big)\right]
\end{aligned}
\tag{E.9}
$$

By Lemma 5, we have

$$
\begin{aligned}
&\mathbb{E}_{x_t \sim \mathcal{D}_\mathcal{X}}\left[\sum_{a_t \in \mathcal{A}} p_{m(t)}(a_t|x)\Big(\mu(x_t, \pi_\mu(x_t)) - \mu(x_t, a_t)\Big)\right. \\
&= \mathbb{E}_{x_t \sim \mathcal{D}_\mathcal{X}}\left[\sum_{a_t \in \mathcal{A}} \sum_{\pi \in \Psi} \mathbb{I}\{\pi(x_t) = a_t\} Q_m(\pi)\Big(\mu(x_t, \pi_\mu(x_t)) - \mu(x_t, a_t)\Big)\right] \\
&= \mathbb{E}_{x_t \sim \mathcal{D}_\mathcal{X}}\left[\sum_{\pi \in \Psi} Q_m(\pi)\Big(\mu(x_t, \pi_\mu(x_t)) - \mu(x_t, \pi(x_t))\Big)\right] \\
&= \sum_{\pi \in \Psi} Q_m(\pi)\mathbb{E}_{x_t \sim \mathcal{D}_\mathcal{X}}\left[\mu(x_t, \pi_\mu(x_t)) - \mu(x_t, \pi(x_t))\right] \\
&= \sum_{\pi \in \Psi} Q_m(\pi)\text{Reg}(\pi)
\end{aligned}
\tag{E.10}
$$

where the last equality is from the definition of the expected regret in the measure $\mu$. □

**Lemma 7.** *Fix any epoch $m \geq m_0 \in \mathbb{N}$ and any round $t$ in epoch $m$, we have:*

$$\sum_{\pi \in \Psi} Q_m(\pi)\widehat{Reg}_t(\pi) < \frac{K}{\gamma_m}. \tag{E.11}$$

*Proof.* For any $t$ in epoch $m$, based on the definition of $\widehat{\text{Reg}}_t(\pi) = \mathbb{E}_{x_t \sim \mathcal{D}_\mathcal{X}}[\widehat{\mu}_{m(t)}(x_t, \pi_{\widehat{\mu}_{m(t)}}) - \widehat{\mu}_{m(t)}(x_t, \pi(x_t))]$ where

$b_t = \pi_{\widehat{\mu}_{m(t)}}(x_t) = \mathrm{argmax}_{i \in [K]} \widehat{\mu}_{m(t)}(x_t, i)$, we have

$$\sum_{\pi \in \Psi} Q_m(\pi) \widehat{\mathrm{Reg}}_t(\pi)$$

$$= \sum_{\pi \in \Psi} Q_m(\pi) \mathbb{E}_{x_t \sim \mathcal{D}_\mathcal{X}} \left[ \widehat{\mu}_{m(t)}(x_t, b_t) - \widehat{\mu}_{m(t)}(x_t, \pi(x_t)) \right]$$

$$= \mathbb{E}_{x_t \sim \mathcal{D}_\mathcal{X}} \left[ \sum_{\pi \in \Psi} Q_m(\pi) \left( \widehat{\mu}_{m(t)}(x_t, b_t) - \widehat{\mu}_{m(t)}(x_t, \pi(x_t)) \right) \right]$$

$$= \mathbb{E}_{x_t \sim \mathcal{D}_\mathcal{X}} \left[ \sum_{a_t \in \mathcal{A}} \sum_{\pi \in \Psi} Q_m(\pi) \mathbb{I}\{\pi(x_t) = a_t\} \left( \widehat{\mu}_{m(t)}(x_t, b_t) - \widehat{\mu}_{m(t)}(x_t, a_t) \right) \right]$$

$$= \mathbb{E}_{x_t \sim \mathcal{D}_\mathcal{X}} \left[ \sum_{a_t \in \mathcal{A}} p_{m(t)}(a_t | x_t) \left( \widehat{\mu}_{m(t)}(x_t, b_t) - \widehat{\mu}_{m(t)}(x_t, a_t) \right) \right] \quad \text{(E.12)}$$

$$= \mathbb{E}_{x_t \sim \mathcal{D}_\mathcal{X}} \left[ \sum_{a_t \in \mathcal{A}} \frac{1}{K + \gamma_m(\widehat{\mu}_{m(t)}(x_t, b_t) - \widehat{\mu}_{m(t)}(x_t, a_t))} \left( \widehat{\mu}_{m(t)}(x_t, b_t) - \widehat{\mu}_{m(t)}(x_t, a_t) \right) \right]$$

$$= \mathbb{E}_{x_t \sim \mathcal{D}_\mathcal{X}} \left[ \sum_{a_t \in \mathcal{A}/\{b_t\}} \frac{1}{\gamma_m} \frac{\gamma_m(\widehat{\mu}_{m(t)}(x_t, b_t) - \widehat{\mu}_{m(t)}(x_t, a_t))}{K + \gamma_m(\widehat{\mu}_{m(t)}(x_t, b_t) - \widehat{\mu}_{m(t)}(x_t, a_t))} \right]$$

$$< \frac{K - 1}{\gamma_m}.$$

where the last inequality holds by the $\frac{\gamma_m(\widehat{\mu}_{m(t)}(x_t, b_t) - \widehat{\mu}_{m(t)}(x_t, a_t))}{K + \gamma_m(\widehat{\mu}_{m(t)}(x_t, b_t) - \widehat{\mu}_{m(t)}(x_t, a_t))} < 1$. $\qquad\square$

The next lemma establishes the relationship between the predicted implicit regret and the true implicit regret of any policy at round $t$. This lemma ensures that the predicted implicit regret of good polices are becoming more and more accurate, while the predicted implicit regret of bad policies do not need to have such property.

**Lemma 8.** *Suppose the event $\Lambda_2$ in Lemma 4 holds, let $C_0 = 204$. For all policies $\pi$ and epoch $m \geq m_0$, we have:*

$$Reg(\pi) \leq 2\widehat{Reg}_t(\pi) + \frac{C_0 K}{\gamma_m}$$

$$\widehat{Reg}_t(\pi) \leq 2Reg(\pi) + \frac{C_0 K}{\gamma_m} \quad \text{(E.13)}$$

That is, for any policy, Lemma 8 bounds the prediction error of the implicit regret estimate.

*Proof.* We prove it via induction on epoch $m$. We first consider the base case when $m = 1$ and $1 \leq t \leq \tau_1$. In this case, since $\gamma_1 = 1$, we know that $\forall \pi \in \Psi$, $Reg(\pi) \leq \sqrt{K} \leq C_0 K/\gamma_1$, $\widehat{Reg}_t(\pi) = 0 \leq C_0 K/\gamma_1$. Note that we use condition $\mathbb{E}_{x \sim \mathcal{P}_X}[\sup_{a, a' \in \mathcal{A}}(\mu(x, a) - \mu(x, a'))] \leq \sqrt{K}$, which is very weak - in the special case of multi-armed bandits, it means "the gap between mean rewards of two actions is no greater than $\sqrt{K}$. Thus the claim holds in the base case.

For the induction step, fix some epoch $m > 1$. We assume that for all epochs $m' \leq m$, all rounds $t'$ in epoch $m'$, and all $\pi \in \Psi$,

$$Reg(\pi) \leq 2\widehat{Reg}_{t'}(\pi) + C_0 \frac{K}{\gamma_{m'}},$$

$$\widehat{Reg}_{t'}(\pi) \leq 2Reg(\pi) + C_0 \frac{K}{\gamma_{m'}}. \quad \text{(E.14)}$$

**Step 1.** For all rounds $t$ in epoch $m$ and all $\pi \in \Psi$, we first show that

$$Reg(\pi) \leq 2\widehat{Reg}_t(\pi) + C_0 \frac{K}{\gamma_m}.$$

Based on the definition of $\text{Reg}(\pi)$ and $\widehat{\text{Reg}}_t$, we have

$$
\begin{aligned}
\text{Reg}(\pi) - \widehat{\text{Reg}}_t(\pi) &= (R_t(\pi_\mu) - R_t(\pi)) - (\widehat{R}_t(\pi_{\widehat{\mu}_{m(t)}}) - \widehat{R}_t(\pi)) \\
&\leq (R_t(\pi_\mu) - R_t(\pi)) - (\widehat{R}_t(\pi_\mu) - \widehat{R}_t(\pi)) \\
&\leq |\widehat{R}_t(\pi) - R_t(\pi)| + |R_t(\pi_\mu) - \widehat{R}_t(\pi_\mu)| \\
&\leq \frac{4\sqrt{\mathcal{V}_t(\pi)}\sqrt{K}}{\gamma_m} + \frac{4\sqrt{\mathcal{V}_t(\pi_\mu)}\sqrt{K}}{\gamma_m} \\
&\leq \frac{\mathcal{V}_t(\pi)}{5\gamma_m} + \frac{\mathcal{V}_t(\pi_\mu)}{5\gamma_m} + \frac{40K}{\gamma_m}
\end{aligned}
\tag{E.15}
$$

where the third inequality holds by Lemma 9, and the last inequality holds by the AM-GM inequality. Based on the definition of $\mathcal{V}_t(\pi), \mathcal{V}_t(\pi_\mu)$ and the upper bound of the expected inverse probability from Lemma 10, there exist epochs at least one $i, j \leq m$ and $t \leq \tau_m$ such that

$$
\mathcal{V}_t(\pi) = V_t(p_i, \pi) = \mathbb{E}_{x_t \sim \mathcal{D}_\mathcal{X}}\left[\frac{1}{p_i(\pi(x_t)|x_t)}\right] \leq K + \gamma_i \widehat{\text{Reg}}_{\tau_i}(\pi)
$$

$$
\mathcal{V}_t(\pi_\mu) = V_t(p_j, \pi_\mu) = \mathbb{E}_{x_t \sim \mathcal{D}_\mathcal{X}}\left[\frac{1}{p_i(\pi_\mu(x_t)|x_t)}\right] \leq K + \gamma_j \widehat{\text{Reg}}_{\tau_j}(\pi_\mu)
$$

Combing above two inequalities with Eq.E.15 of induction and $\gamma_i, \gamma_j \leq \gamma_m$, we have

$$
\frac{\mathcal{V}_t(\pi)}{5\gamma_m} \leq \frac{K + \gamma_i \widehat{\text{Reg}}_{\tau_i}(\pi)}{5\gamma_m} \leq \frac{K + \gamma_i(2\text{Reg}(\pi) + C_0\frac{K}{\gamma_i})}{5\gamma_m} \leq \frac{(1 + C_0)K}{5\gamma_m} + \frac{2}{5}\text{Reg}(\pi)
$$

$$
\frac{\mathcal{V}_t(\pi_\mu)}{5\gamma_m} \leq \frac{K + \gamma_j \widehat{\text{Reg}}_{\tau_j}(\pi_\mu)}{5\gamma_m} \leq \frac{K + \gamma_j(2\text{Reg}(\pi_\mu) + C_0\frac{K}{\gamma_j})}{5\gamma_m} = \frac{(1 + C_0)K}{5\gamma_m}
$$

where the last equality by $\text{Reg}(\pi_\mu) = 0$. Combining all above, we have

$$
\text{Reg}(\pi) - \widehat{\text{Reg}}_t(\pi) \leq \frac{2}{5}\text{Reg}(\pi) + \frac{2(1 + C_0)K}{5\gamma_m} + \frac{40K}{\gamma_m}
$$

which is equivalent to

$$
\text{Reg}(\pi) \leq \frac{5}{3}\widehat{\text{Reg}}_t(\pi) + \frac{2C_0 K}{3\gamma_m} + \frac{68K}{\gamma_m} \leq 2\widehat{\text{Reg}}_t(\pi) + \frac{C_0 K}{\gamma_m},
$$

by $C_0 \leq 204$.

**Step 2.** We then show for all rounds $t$ in epoch $m$ and all $\pi \in \Psi$,

$$
\widehat{\text{Reg}}_t(\pi) \leq 2\text{Reg}(\pi) + \frac{C_0 K}{\gamma_m}.
\tag{E.16}
$$

Similar to step 2, we can get the similar result. Thus we complete the inductive step, and the claim proves to be true for all $m \in \mathbb{N}$. $\qquad\square$

This following lemma is a key step to provide the relationship of $\widehat{\text{Reg}}_t(\pi)$ and $\text{Reg}(\pi)$ in Lemma 8.

**Lemma 9.** *For any round $t \geq \tau_{m_0} + 1$, for any policy $\pi \in \Psi$, we have*

$$
|\widehat{R}_t(\pi) - R_t(\pi)| \leq \frac{4\sqrt{\mathcal{V}_t(\pi)}\sqrt{K}}{\gamma_{m(t)}}
\tag{E.17}
$$

*Proof.* Fix any policy $\pi \in \Psi$, and any round $t > \tau_{m_0-1}$. By the definition of $\widehat{R}_t(\pi)$ and $R_t(\pi)$, we have

$$
\widehat{R}_t(\pi) - R_t(\pi) = \mathbb{E}_{x_t \sim \mathcal{D}_\mathcal{X}}\left[\widehat{\mu}_{m(t)}(x_t, \pi(x_t)) - \mu(x_t, \pi(x_t))\right]
\tag{E.18}
$$

Given a context $x_t$, define $\Delta_{x_t} = \widehat{\mu}_{m(t)}(x_t, \pi(x_t)) - \mu(x_t, \pi(x_t))$, then we have the equality $\mathbb{E}_{x_t \sim \mathcal{D}_\mathcal{X}}[\Delta_x] = \widehat{R}_t(\pi) - R_t(\pi)$. For all $s = \tau_{m_0 - 1} + 1, ..., \tau_{m(t)-1}$, we have

$$
\mathbb{E}_{a_s|x_s}\left[\left(\widehat{\mu}_{m(t)}(x_s, a_s) - \mu(x_s, a_s)\right)^2 | \mathfrak{S}_{t-1}\right]
$$
$$
= \sum_{a_s \in \mathcal{A}} p_{m(s)}(a_s|x_s)\left[\widehat{\mu}_{m(t)}(x_s, a_s) - \mu(x_s, a_s)\right]^2 \tag{E.19}
$$
$$
\geq p_{m(s)}(\pi(x_s)|x_s)\left[\widehat{\mu}_{m(t)}(x_s, \pi(x_s)) - \mu(x_s, \pi(x_s))\right]^2
$$
$$
= p_{m(s)}(\pi(x_s)|x_s)\Delta_{x_s}^2
$$

where the first inequality holds by the kernel and squared terms both positive and ignoring other actions $a_s \neq \pi(x_s)$. Then we can take a sum of regret difference over the epoch $m$ and multiply it by the maximum expected inverse probability $\mathcal{V}_t(\pi)$, defined in Eq.E.3. For the start of the epoch $m(t)$, we define $s_0 = \tau_{m(t)-1} + 1$ and assume $m(t) > m_0$, we have

$$
\mathcal{V}_t(\pi) \sum_{s=s_0}^{\tau_{m(t)}-1} \mathbb{E}_{x_s, a_s}\left[\left(\widehat{\mu}_{m(t)}(x_s, a_s) - \mu(x_s, a_s)\right)^2 | \mathfrak{S}_{t-1}\right]
$$
$$
\geq \sum_{s=s_0}^{\tau_{m(t)}-1} V(p_{m(s)}, \pi)\mathbb{E}_{x_s, a_s}\left[\left(\widehat{\mu}_{m(t)}(x_s, a_s) - \mu(x_s, a_s)\right)^2 | \mathfrak{S}_{t-1}\right]
$$
$$
= \sum_{s=s_0}^{\tau_{m(t)}-1} \mathbb{E}_{x_s}\left[\frac{1}{p_{m(s)}(\pi(x_s)|x_s)}\right] \mathbb{E}_{x_s}\mathbb{E}_{a_s|x_s}\left[\left(\widehat{\mu}_{m(t)}(x_s, a_s) - \mu(x_s, a_s)\right)^2 | \mathfrak{S}_{t-1}\right] \tag{E.20}
$$
$$
\geq \sum_{s=s_0}^{\tau_{m(t)}-1} \left(\mathbb{E}_{x_s}\left[\sqrt{\frac{1}{p_{m(s)}(\pi(x_s)|x_s)}\mathbb{E}_{a_s|x_s}\left[\left(\widehat{\mu}_{m(t)}(x_s, a_s) - \mu(x_s, a_s)\right)^2 | \mathfrak{S}_{t-1}\right]}\right]\right)^2
$$

where the first inequality from the definition of $\mathcal{V}_t(\pi)$ and the second follows the Cauchy-Schwarz inequality. By the above inequality from Eq.E.19, we have the following

$$
\geq \sum_{s=s_0}^{\tau_{m(t)}-1} \left(\mathbb{E}_{x_s}\left[\sqrt{\frac{1}{p_{m(s)}(\pi(x_s)|x_s)}p_{m(s)}(\pi(x_s)|x_s)\Delta_{x_s}^2}\right]\right)^2
$$
$$
= \sum_{s=s_0}^{\tau_{m(t)}-1} \left(\mathbb{E}_{x_s}[|\Delta_{x_s}|]\right)^2 \tag{E.21}
$$
$$
\geq \sum_{s=s_0}^{\tau_{m(t)}-1} |\widehat{R}_t(\pi) - R_t(\pi)|^2
$$
$$
= (\tau_{m(t)} - s_0)|\widehat{R}_t(\pi) - R_t(\pi)|^2
$$

and the last inequality follows from the convexity of the $l_1$ norm and last equality holds by the definition of $\widehat{R}_t(\pi)$ and $R_t(\pi)$. So we have

$$
|\widehat{R}_t(\pi) - R_t(\pi)| \leq \sqrt{\mathcal{V}_t(\pi)}\sqrt{\frac{\sum_{s=s_0}^{\tau_{m(t)}-1} \mathbb{E}_{x_s, a_s}\left[\left(\widehat{\mu}_{m(t)}(x_s, a_s) - \mu(x_s, a_s)\right)^2 | \mathfrak{S}_{t-1}\right]}{\tau_{m(t)} - \tau_{m(t)-1}}}
$$
$$
\leq \frac{4\sqrt{\mathcal{V}_t(\pi)}\sqrt{K}}{\gamma_{m(t)}} \tag{E.22}
$$

where the last inequality holds by the definition of the exploitation rate of $\gamma_{m(t)}$. $\qquad\square$

The following Lemma is a key step to control the expected inverse probability $V(p_{m(t)}, \pi)$.

**Lemma 10.** *Fix any epoch $m \geq m_0 \in \mathbb{N}$, we have:*

$$V(p_{m(t)}, \pi) \leq K + \gamma_m \widehat{Reg}_t(\pi) \tag{E.23}$$

*Proof.* For any policy $\pi \in \Psi$, given any context $x_t \in \mathcal{X}$, we have

$$\frac{1}{p_{m(t)}(\pi(x_t)|x_t)} \begin{cases} = K + \gamma_m(\widehat{\mu}_{m(t)}(x_t, b_t) - \widehat{\mu}_{m(t)}(x_t, \pi(x_t))), & \text{if } \pi(x_t) \neq b_t; \\ \leq \frac{1}{1/K} = K = K + \gamma_m(\widehat{\mu}_{m(t)}(x_t, b_t) - \widehat{\mu}_{m(t)}(x_t, \pi(x_t))), & \text{if } \pi(x_t) = b_t. \end{cases} \tag{E.24}$$

Based on the definition of the expected inverse probability in Eq.E.2, we have

$$\begin{aligned} V(p_{m(t)}, \pi) &= \mathbb{E}_{x_t \sim \mathcal{D}_{\mathcal{X}}} \left[ \frac{1}{p_{m(t)}(\pi(x_t)|x_t)} \right] \\ &\leq K + \gamma_m \mathbb{E}_{x_t \sim \mathcal{D}_{\mathcal{X}}} \left[ \widehat{\mu}_{m(t)}(x_t, b_t) - \widehat{\mu}_{m(t)}(x_t, \pi(x_t)) \right] \\ &= K + \gamma_m \widehat{Reg}_t(\pi), \end{aligned} \tag{E.25}$$

where the inequality follows by the condition if $\pi(x_t) = b_t$ and the last equation is followed by the definition of the expected regret in the measure of empirical $\widehat{\mu}_{m(t)}$. $\square$

The following lemma provides the key step to provide the regret upper bound.

**Lemma 11.** *For any $T \in \mathbb{N}$, wiht probability at least $1 - \delta$, the expected regret of RCB after $T$ rounds is at most* $\tau_{m_0-1} + 206K \sum_{t=\tau_{m_0-1}+1}^{T} 1/\gamma_{m(t)} + \sqrt{8(T - \tau_{m_0-1}) \log(2/\delta)}$.

*Proof.* For each round $t \geq \tau_{m_0-1} + 1$, define $M_t := y_t(\pi_\mu) - y_t(a_t) - \sum_{\pi \in \Psi} Q_m(\pi) \text{Reg}(\pi)$ and $M_t$ is a martingale difference sequence since $\mathbb{E}_{x_t,y_t,a_t}[M_t|\mathfrak{S}_{t-1}] = 0$ provided by Lemma 6. So we have

$$\mathbb{E}_{x_t,y_t,a_t} \left[ y_t(\pi_\mu) - y_t(a_t)|\mathfrak{S}_{t-1} \right] = \sum_{\pi \in \Psi} Q_m(\pi) \text{Reg}(\pi), \tag{E.26}$$

Since $|M_t| \leq 2$ by $y_t \in [0, 1]$, by the Azuma-Hoeffding's inequality from Lemma 3,

$$\sum_{t=\tau_{m_0-1}+1}^{T} M_t \leq \sqrt{8(T - \tau_{m_0-1}) \log(\frac{2}{\delta})} \tag{E.27}$$

with probability at least $1 - \delta/2$. By Lemma 4, we can upper bound the regret the with probability at least $1 - \delta/2$,

$$\begin{aligned} &\sum_{t=\tau_{m_0-1}+1}^{T} \mathbb{E} \left[ y_t(\pi_\mu) - y_t(a_t)|\mathfrak{S}_{t-1} \right] \\ &\leq \sum_{t=\tau_{m_0-1}+1}^{T} \sum_{\pi \in \Psi} Q_m(\pi) \text{Reg}(\pi) + \sqrt{8(T - \tau_{m_0-1}) \log(\frac{2}{\delta})} \\ &\leq \sum_{t=\tau_{m_0-1}+1}^{T} \sum_{\pi \in \Psi} Q_m(\pi)(2\widehat{\text{Reg}}_t(\pi) + \frac{C_0 K}{\gamma_m}) + \sqrt{8(T - \tau_{m_0-1}) \log(\frac{2}{\delta})} \\ &= \sum_{t=\tau_{m_0-1}+1}^{T} \sum_{\pi \in \Psi} [2Q_m(\pi)\widehat{\text{Reg}}_t(\pi) + Q_m(\pi)\frac{C_0 K}{\gamma_m}] + \sqrt{8(T - \tau_{m_0-1}) \log(\frac{2}{\delta})} \\ &\leq \sum_{t=\tau_{m_0-1}+1}^{T} [\frac{2K}{\gamma_m} + \frac{C_0 K}{\gamma_m}] + \sqrt{8(T - \tau_{m_0-1}) \log(\frac{2}{\delta})} \\ &\leq 206 \sum_{t=\tau_{m_0-1}+1}^{T} \frac{K}{\gamma_{m(t)}} + \sqrt{8(T - \tau_{m_0-1}) \log(\frac{2}{\delta})} \end{aligned} \tag{E.28}$$

where the second inequality holds by Lemma 8 to control the implicit expected regret in $\pi$ and the third inequality holds by Lemma 7 controlling the empirical regret. $\qquad\square$

### E.2. Proof of Theorem 2

*Proof.* By Lemma 11, with probability $1 - \delta$, we have

$$\sum_{t=1}^{T} \mathbb{E}_{x_t \sim \mathcal{D}_\mathcal{X}} (y_t(\pi_\mu) - y_t(a_t))$$

$$\leq \tau_{m_0-1} + \sum_{t=\tau_{m_0-1}+1}^{T} \frac{206K}{\gamma_{m(t)}} + \sqrt{8(T - \tau_{m_0-1}) \log(\frac{2}{\delta})} \tag{E.29}$$

$$\leq \tau_{m_0-1} + 52 \sum_{m=m_0}^{m_1} \sqrt{K\mathcal{E}_\mathcal{F}(\tau_{m-2}, \tau_{m-1})(\tau_m - \tau_{m-1})} + \sqrt{8(T - \tau_{m_0-1}) \log(\frac{2}{\delta})}.$$

With the assumption that the prior distribution $\mathcal{P}_0$ is normal and the variance is increasing in order $\mathcal{O}(t)$, by $\tau_m = 2^m$, we have

$$= \tau_{m_0-1} + 52\sigma\sqrt{Kd} \sum_{m=m_0}^{m_1} \mathcal{O}(\frac{1}{\sqrt{2^{m-2}}})2^{m-1} + \sqrt{8(T - \tau_{m_0-1}) \log(\frac{2}{\delta})}$$

$$= \tau_{m_0-1} + 52\sigma\sqrt{Kd} \sum_{m=m_0}^{m_1} \mathcal{O}(\sqrt{2^m}) + \sqrt{8(T - \tau_{m_0-1}) \log(\frac{2}{\delta})}$$

$$\leq \tau_{m_0-1} + 52\sigma\sqrt{Kd} \int_{m_0}^{\log_2(T)} 2^{\frac{x}{2}} dx + \sqrt{8(T - \tau_{m_0-1}) \log(\frac{2}{\delta})} \tag{E.30}$$

$$\leq \tau_{m_0-1} + \frac{104}{\ln 2}\sigma\sqrt{KdT} + \sqrt{8(T - \tau_{m_0-1}) \log(\frac{2}{\delta})}$$

$$< \tau_{m_0-1} + 151\sigma\sqrt{KdT} + \sqrt{8(T - \tau_{m_0-1}) \log(\frac{2}{\delta})}$$

$$\square$$

## F. Additional Experiments Results

### F.1. Simulation Studies

The goal of this section is to demonstrate that RCB algorithm can satisfy the DBIC constraint and simutaneously secure the sublinear regret. For all settings, the following parameters need to be specified (a) *environment parameters*: time horizon $T$, number of arms $K$, feature dimension $d$, and noise level $\sigma$; (b) *DBIC parameters*: budget $\epsilon$, prior-posterior minimum gap constants $\tau_{\mathcal{P}_0}$ and $\rho_{\mathcal{P}_0}$; (c) *prior belief parameters*: prior $\mathcal{P}_0$, where we assume the prior follows the normal distribution.

**Setting 1 (Environment Effects)**: We consider RCB's robustness in terms of different $K = [2, 5, 10]$, $d = [3, 5, 10]$. For rest parameters, we set $T = 10^5$, $\sigma = 0.05$, $\epsilon = 0.05$, $\tau_{\mathcal{P}_0} = 0.01$, and $\rho_{\mathcal{P}_0} = 0.95$. The prior are set to be $\beta_{i,0} = \mathbf{0}_d$ and $\Sigma_{i,0} = 1/5\mathbf{I}_d$.

**Setting 2 (Ad-hoc Design)**: This scenario demonstrates the results when the platform adopts an ad-hoc approach to $\mathsf{N}(\epsilon)$ without following the guidelines of Theorem 1. Here, $\mathsf{N}$ is set to $\{10, 100, 1000\}$. All other parameters remain consistent with those specified in Setting 1.

### F.2. Discussion of Setting 1 and Setting 2

**Analysis of Setting 1 (Upper part of Figure 3)**: Different columns in the figure represent various dimensions $d$, with the first three columns illustrating the DBIC gain and the last three columns detailing the regrets observed. Our findings indicate that RCB satisfies the DBIC property, as evidenced by the gain consistently exceeding -0.05 (dashed line), or budget not been used up. During the Exploitation stage, there is an observable upward trend in the instantaneous DBIC gain, suggesting that the recommendation system increasingly gains trust from customers (larger $\epsilon$ gain). The right segment of the figure explores the relationship between regret, $d$, and $K$. It was observed that the regret for $K = 10$ significantly exceeds that for

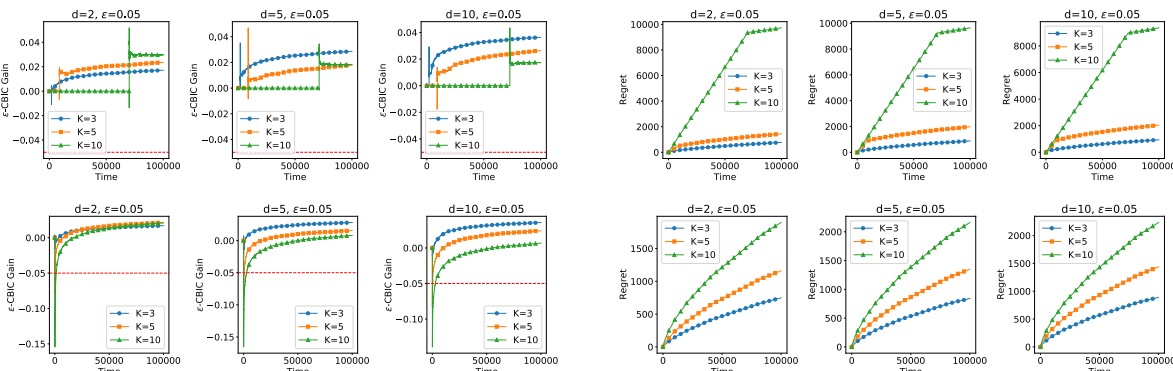

*Figure 3.* Incentive gain (left) and cumulative regret (right) of Setting 1 (upper) and Setting 2 (lower).

$K = 3$ and $K = 5$. This discrepancy arises because, to maintain the DBIC property, the duration of the cold start stage increases cubically with $K$, representing a substantial cost during this initial phase. In contrast, the impact of $d$ on cost is relatively minimal, as articulated in Theorem 1.

**Analysis of Setting 2 (Lower part of Figure 3):** This setting mirrors Setting 1 in terms of overall configuration. However, in this scenario, the platform does not adhere to the sample size requirements needed to satisfy the DBIC property, opting instead for an arbitrary fixed cold start length of $\mathsf{N}(\epsilon) = \{10, 100, 1000\}$. The simulation results for $\mathsf{N}(\epsilon) = \{100, 1000\}$ are detailed in Appendix §F. When compared with the regret observed in Setting 1, which is at the level of $10^5$, the regret in Setting 2 is considerably lower, at approximately $10^3$. However, in terms of DBIC gain, Setting 1 consistently shows positive gains, fully complying with the DBIC property, whereas Setting 2 experiences periods of negative gains, particularly when the number of arms is high ($K = 10$). This negative trend is more pronounced as $d$ increases, making it increasingly challenging to estimate an appropriate cold start length, as further discussed in Appendix §F. Notably, even with $\mathsf{N}(\epsilon) = 1000$, the DBIC gain remains negative for most instances when $d = 5$ or 10.

### F.3. Additional Simulation Settings and Results Analysis

**Setting 3 ($\epsilon$ effects):** We consider the RCB algorithm's effect over different budget parameters with $\epsilon = [0.01, 0.03, 0.05]$ and prior variances $\Sigma_{i,0} = 1/\lambda \mathbf{I}_d = [1/3, 1/5, 1/10]\mathbf{I}_d$. For rest parameters, $T = 5 \times 10^4$, $K = 5$, $d = 5$, $\sigma = 0.05$, and $\beta_{i,0} = \mathbf{0}_d, \forall i \in [K]$.

**Setting 4 (Prior Decay and Prior-Posterior Gap Assumption Mis-specification Effects):** We also test the robustness of RCB algorithm when the Assumption 4 is mis-specified. Here we assume $\Sigma_{i,0} = [0.02, 0.04, 0.1]\mathbf{I}$ and the prior decay rate are *linear decay, square root decay, and log decay*. We set the environment parameters to be $T = 5^4$, $K = 5$, $d = 5$. We set $\epsilon = 0.05$ and the prior mean $\beta_{1,0} = [1, 1, 1, 1, 1]^\mathsf{T}$ and $\beta_{i,0} = [0, 0, 0, 0, 0]^\mathsf{T}$.

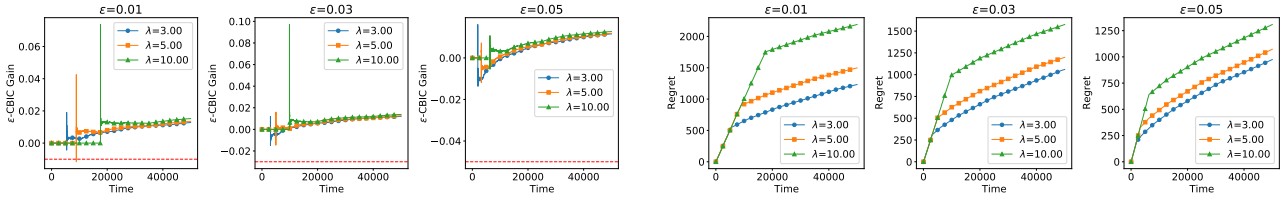

*Figure 4.* Gain (top) and Regret (bottom) of Setting 2.

**Analysis of Setting 1 (Upper part of Figure 3):** Different columns in the figure represent various dimensions $d$, with the first three columns illustrating the DBIC gain and the last three columns detailing the regrets observed. Our findings indicate that RCB satisfies the DBIC property, as evidenced by the gain consistently exceeding -0.05 (dashed line), or budget not been used up. During the Exploitation stage, there is an observable upward trend in the instantaneous DBIC gain, suggesting that the recommendation system increasingly gains trust from customers (larger $\epsilon$ gain). The right segment of the figure explores the relationship between regret, $d$, and $K$. It was observed that the regret for $K = 10$ significantly exceeds that for

$K = 3$ and $K = 5$. This discrepancy arises because, to maintain the DBIC property, the duration of the cold start stage increases cubically with $K$, representing a substantial cost during this initial phase. In contrast, the impact of $d$ on cost is relatively minimal, as articulated in Theorem 1.

**Analysis of Setting 2 (Lower part of Figure 3)**: This setting mirrors Setting 1 in terms of overall configuration. However, in this scenario, the platform does not adhere to the sample size requirements needed to satisfy the DBIC property, opting instead for an arbitrary fixed cold start length of $\mathsf{N}(\epsilon) = \{10, 100, 1000\}$. The simulation results for $\mathsf{N}(\epsilon) = \{100, 1000\}$ are detailed in Appendix §F. When compared with the regret observed in Setting 1, which is at the level of $10^5$, the regret in Setting 2 is considerably lower, at approximately $10^3$. However, in terms of DBIC gain, Setting 1 consistently shows positive gains, fully complying with the DBIC property, whereas Setting 2 experiences periods of negative gains, particularly when the number of arms is high ($K = 10$). This negative trend is more pronounced as $d$ increases, making it increasingly challenging to estimate an appropriate cold start length, as further discussed in Appendix §F. Notably, even with $\mathsf{N}(\epsilon) = 1000$, the DBIC gain remains negative for most instances when $d = 5$ or 10.

**Setting 3 - $\epsilon$ Effects Analysis:** In Figure 4, three columns represent different $\epsilon$'s effects over the DBIC gain and regret.

For the top of the figure, we found that `RCB` can satisfy the DBIC property under different $\epsilon$ and $\lambda$'s scenario. What's more, all the instantaneous gains have the uplift trend (increasing gain), which shows similar pattern to the setting 1.

The bottom shows the relationship between the regret, $\epsilon$, and the prior variance $\Sigma_{i,0} = 1/\lambda \mathbf{I}_d$. We found that the regret of $\Sigma_{i,0} = 1/10\mathbf{I}_d$ is much larger than the regret of $\Sigma_{i,0} = 1/3\mathbf{I}_d$ and $\Sigma_{i,0} = 1/5\mathbf{I}_d$. The reason is that in order to satisfy DBIC property, the length of the cold start stage is linearly inverse proportion to the order of minimum eigenvalue $\phi_0$, which is demonstrated in Theorem 1. In other words, when the prior variance is small, it means that the customers have strong opinions over arms and the platform needs a long length of the cold start stage to make the `RCB` algorithm to satisfy the DBIC property. In addition, the regret will decreases when $\epsilon$ increases. That is, when the platform wants to avoid long length of the cold start stage, it can sacrifice the $\epsilon$ to avoid a large regret, which is a trade-off between the guarantee of DBIC property and the regret.

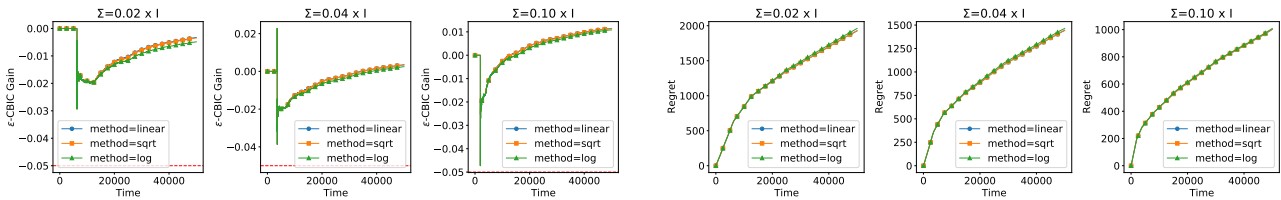

*Figure 5.* Gain (top) and Regret (bottom) of Setting 2.

**Setting 4 - Misspecified Effects Analysis:** In Figure 5, the three columns represent different prior margin $\tau_{\mathcal{P}_0}$'s effects over the regret and decay rate mis-specified over the DBIC gain. For top figure, we found `RCB` can still protect the DBIC under different $\Sigma$ scenario. Besides, we found that all the instantaneous DBIC gains still have the uplift trend, which shows similar pattern to the setting 1 and setting 2. And the linear decay rate has the largest DBIC gain and as $\Sigma_{i,0}$ increases, the platform gains more.

The second row shows the relationship between the regret and margin, and the decay rate misspecified. We found that in any decay rate that the `RCB` algorithm employs, the regret of are really similar. The reason is that for any element of $\beta_{i,0}$ is small within $[0, 1]$ and the prior variance is moderate, three decay rates has similar effect. And we found that when variance increases, regret decrease. It indicates that when prior variance is large, the regret difference among three different decay rates is shrinkage. In other words, when costumers do not have strong opinions over arms (variance is large), different decay rates have similar regret effects.

### F.4. Additional Real Data Analysis
**Data Description:** This data contains the true patient-specific optimal warfarin doses (which are initially unknown but are eventually found through the physician-guided dose adjustment process over the course of a few weeks) for 5528 patients with more than 70 features. It also includes patient-level covariates such as clinical factors, demographic variables, and

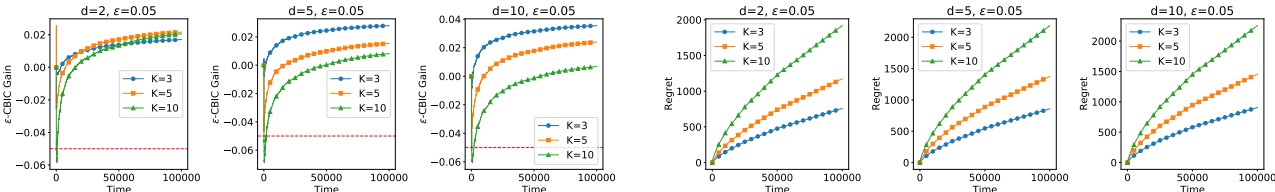

*Figure 6.* Gain (top) and Regret (bottom) of Setting 2 with $N = 10^2$.

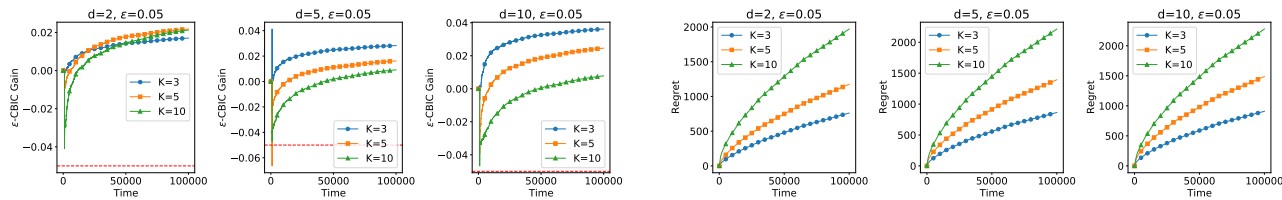

*Figure 7.* Gain (top) and Regret (bottom) of Setting 2 with $N = 10^3$.

genetic information that have been found to be predictive of the optimal warfarin dosage (Consortium, 2009). We follow the similar data construction method in (Bastani & Bayati, 2020). These covariates include:

- Demographics: gender, race, ethnicity, age, height (cm), weight (kg).

- Diagnosis: reason for treatment (e.g. deep vein thrombosis, pulmonary embolism, etc.).

- Pre-existing diagnoses: indicators for diabetes, congestive heart failure or cardiomyopathy, valve replacement, smoker status.

- Medications: indicators for potentially interacting drugs (aspirin, Tylenol, and Zocor).

- Genetics: presence of genotype variants of CYP2C9 and VKORC1.

The details can be found in Appendix 1 of (Consortium, 2009). All these covariates were hand-selected by professionals as being relevant to the task of warfarin dosing based on medical literature; there are no extraneously added variables. Since the detailed feature construction is not available in (Bastani & Bayati, 2020), we construct features follow the description in (Bastani & Bayati, 2020). For *diagnosis variables*, we categorize the reason for treatment with 0/1 (1 represents patients have reason for treatment, 0 represents patients have no reason or unknown reason for treatment). For *medications variables*, we only include three medications: aspirin, Tylenol, Zocor, and all other medications are set to be 0. For *genetics variables*, we considered genotype variants of CYP2C9 and VKORC1 and the rest are set to be 0. The previous feature construction aims to avoid to high dimensional feature space. All categorical variables are transformed into dummy variables and all missing values are set to 0. After the data construction, we have 70 features and 5528 patients. In (Bastani & Bayati, 2020), they have 93 features, which is similar to our constructions.

**Model Hyperparameter Setup:** The prior mean's setup follow the fixed-dose strategy and detailed explanation is provided in the following. We assume the prior variance increases linearly over time after the *cold start*. This allows physicians decease the confidence of their prior dose strategy and trust the RCB algorithm over time. In addition, the length of the *cold start* is determined by Theorem 1.

**Addition Result Analysis.**

**Regret:** In the first row, we show the regret of RCB with different confidence strengths (prior variance). When $\Sigma$ is small that means physicians have stronger opinion over the medium dosage, and the reverse is that the physicians have weaker opinion over the medium dosage. With different prior, we found that when $\Sigma = 0.4\mathbf{I}$, it has the largest regret since we need more samples in the cold start stage to let physicians trust RCB, which means that we need a large $N$. Interestingly, we found that when $\epsilon$ increases (left to right), the regret difference between different prior variance shrinks because when we can

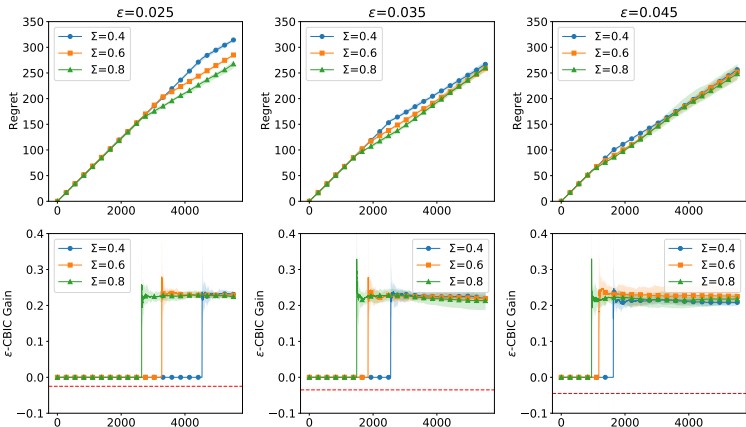

*Figure 8.* Regret and incentive compatibility of warfarin dosing.

tolerate with a higher ratio of non-DBIC compatible patients, the prior's effect decreases and the overall regret decreases because of a shorter cold start stage.

$\epsilon$-**DBICGain:** In the second row, we show DBIC gain of the `RCB` with different confidence strengths. Different prior variance has similar effect on the DBICgain and all variants' gain are above $-\epsilon$, which satisfies the property since the gain after the cold start stage is only determined by the posterior difference within the arm `RCB` selected.

## G. Nonlinear Reward Discussion

If the true model has a non-linear structure, we can approximate the nonlinear functions of the covariates by using basis expansion methods in from statistical learning (Hastie et al., 2009).

