[1]Anonymous Institution, Anonymous City, Anonymous Region, Anonymous Country. Correspondence to: Anonymous Author <anon.email@domain.com>.

Preliminary work. Under review by the International Conference on Machine Learning (ICML). Do not distribute.

Table 1: Comparisons between our work and previous works.

| Feature | Mansour et al. (2020) | Sellke (2023) | Our Work (RCB) |
|---|---|---|---|
| **Problem Setting** | Multi-Armed Bandit (MAB) | Linear Contextual Bandit | Linear Contextual Bandit |
| **Context Model** | None or Black-box reduction (ignores linear structure) | **Fixed Design** (Features are static/owned by products) | **Stochastic Covariates** (User features sampled online) |
| **Mechanism** | Hidden Exploration (Phases) | Thompson Sampling | Two-Stage (Cold Start + Gap Sampling) |
| **Key Gap Filled** | Establishes BIC for MAB | Analyzes linear rewards under fixed contexts | Handles **dynamic user contexts** where the best arm changes per round |

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

Assumption 1 guarantees that the prior distribution of an unexplored arm is not *stochastically dominated* by the posterior distributions of the saturated arms. This ensures the

platform can provide a "fighting chance" for cold-start products: even if an arm $i$ has a low average prior mean, there remains a positive probability $\rho_{P0}$ that, for a specific user context $x_t$ and history realization, arm $i$ exceeds the utility of the safe arms by a margin of at least $\tau_{P0}$. Otherwise, it is a trivial problem.

Additionally, we extend this non-degeneracy to the *Exploitation Stage* (where all arms have $N$ samples). We assume that after collecting $N_{P*}$ samples, the gap between the optimal arm and the suboptimal arms is distinguishable. Specifically, the posterior gap exceeds $\tau_{P*}$ with probability at least $\rho_{P*}$. In practice, the constants $\tau_{P0}$ and $\tau_{P*}$ serve as hyperparameters regulating the exploration aggressiveness.

**Assumption 2** (Posterior Distribution Assumption). *Denote $G_t(b_t) = \min_{j \neq b_t} \mathbb{E}[\mu(x_t, b_t) - \mu(x_t, j)|S]$ as the minimum posterior gap when we have N samples of each arms in the Exploitation stage. There exist a uniform time-independent posterior constants $n_{\mathcal{P}_*}, \tau_{\mathcal{P}_*}, \rho_{\mathcal{P}_*} > 0$ such that $\forall n \geq n_{\mathcal{P}_*}, i \in [K]$, then $Pr(G_t(b_t) \geq \tau_{\mathcal{P}_*}) \geq \rho_{\mathcal{P}_*}$.*

Then we provide the regularity conditions over covariates $\mathcal{P}_X$ as follows to avoid the singularity.

**Assumption 3** (Minimum Eigenvalue of $\Sigma$). *Define the minimum eigenvalue of the covariance matrix of $X$ as $\lambda_{\min}(\Sigma) = \lambda_{\min}(\mathbb{E}_{x \sim \mathcal{P}_X}[xx^{\mathsf{T}}])$. There exists such a $\phi_0 > 0$ satisfying that $\lambda_{\min}(\Sigma) \geq \phi_0$.*

**Assumption 4** (Evolution of Trust). *We posit that user priors are not static but evolve to become more diffuse over time. Specifically, we assume the minimum eigenvalues of the prior covariance matrix $\Sigma_{i,0}$ increase with order $\mathcal{O}(t)$.*

This models the diminishing reliance on intrinsic bias. Initially, users may hold strong, specific beliefs (tight priors) about products. However, as they interact with the system, we assume a behavioral shift where users become less "dogmatic" and more open to the platform's signals. Mathematically, expanding the covariance matrix reduces the weight of the prior relative to the likelihood in the posterior update. This lowers the "persuasion threshold" required to satisfy the constraint, reflecting a state where users are increasingly willing to test recommendations that deviate from their initial myopic preferences. We explicitly analyze the robustness of our algorithm when this assumption is violated in Appendix §F.