# OpenReview forum: "Incentivized Exploration with Stochastic Covariates: A Two-Stage Mechanism Design for Recommender System"
_ICML.cc/2026/Conference — ICML 2026 regular_

### Official Review · Reviewer_ee1x · 2026-03-04

**Soundness:** 3
**Presentation:** 3
**Significance:** 3
**Originality:** 3
**Overall Recommendation:** 5
**Confidence:** 3

**Summary:**

This manuscript addresses the concept of incentivized exploration in contextual bandits with stochastic covariates, a key challenge in recommender systems where self-interested users may reject exploratory recommendations. The study appears to outline the concept of a two-stage algorithm (RCB) that first collects a minimum sample size to satisfy Dynamic Bayesian Incentive Compatibility (DBIC), then employs an inverse proportional gap sampling strategy for efficient exploitation. Theoretically, it achieves $\tilde{O}(\sqrt{KdT})$ regret and quantifies the trade-off between incentive budget and exploration cost. Empirical validation on a warfarin dosing case study demonstrates improved decision accuracy for high-risk patient subgroups compared to a fixed clinical baseline.

**Compliance With Llm Reviewing Policy:**

Affirmed.

**Final Justification:**

I have read the responses and keep the original score.

**Key Questions For Authors:**

- How would the algorithm perform if the key assumption of "evolving trust" (Assumption 4) is violated, i.e., if user priors remain stubborn or change unpredictably?
- Could the modular exploitation stage integrate a non-linear offline oracle (e.g., a neural network) while still maintaining the theoretical DBIC and regret guarantees?
- The cold-start sample complexity scales with $K^3$. For a large number of arms (products), is this cost prohibitive, and are there practical strategies to mitigate it?

**Limitations:**

yes

**Strengths And Weaknesses:**

## Pros
- The work tackles a practically significant and theoretically grounded challenge—aligning long-term platform learning with users' immediate self-interest—through the formal lens of DBIC.
- The proposed two-stage RCB framework is clearly presented, with a modular architecture that decouples incentive satisfaction (cold-start) from efficient learning (exploitation), enhancing interpretability and potential for extension.
- The application to personalized warfarin dosing provides compelling, real-world evidence of the framework's utility in a high-stakes domain, effectively quantifying performance gains over a fixed policy.
## Cons
- The DBIC constraint and user decision model assume myopic, rational users with a stable incentive budget $\epsilon$. This may not fully capture real-world user behavior, which can be adaptive, heterogeneous in rationality, or influenced by factors beyond immediate expected reward.
- The theoretical guarantees rely on several non-trivial assumptions (e.g., non-degenerate priors, evolving user trust per Assumption 4, Gaussian priors for sample complexity). The practical validity and robustness of these assumptions, especially in complex recommendation environments, require further discussion.
- While outperforming a fixed policy, the empirical comparison lacks benchmarks against state-of-the-art bandit or incentivized exploration methods on the same real-world dataset, making it difficult to gauge the relative advancement.

---

> ### Author Rebuttal · Authors · 2026-03-28
>
> We thank the reviewer for the thorough and positive assessment. We address each question below.
>
> __Q1: What If "Evolving Trust" (Assumption 4) Is Violated?__
>
> __A1__. If user priors remain stubborn (i.e., users do not update their beliefs based on observed outcomes), then DBIC exploration becomes fundamentally harder — the mechanism cannot build credibility regardless of how much data it collects. Here we can split this scenario in two cases, only a few users violate this or all users violate it. Let's consider all users violating it, a much more difficult case.
>
> __Stage 1__ still works because it relies on mixing exploration with myopic-optimal recommendations, which satisfies BIC regardless of trust evolution.
>
> __Stage 2__ may require modification:
>
> - If trust evolves _slowly_ (but not zero), the epoch lengths in Stage 2 would need to increase, leading to higher regret but the same asymptotic order.
> - If trust is completely frozen, the mechanism degenerates to the Stage 1 strategy throughout, giving $O(T)$ regret — but this is information-theoretically unavoidable since the user never updates beliefs.
>
> In practice, Assumption 4 is mild: it only requires that users' posteriors converge as data accumulates, which follows from standard Bayesian consistency as long as the prior has full support.
>
> We have explicitly analyzed robustness when this assumption is violated in Appendix §F. Practically, a violation means the platform would require a longer Cold Start phase (larger $N(\epsilon)$) or a looser incentive budget to maintain user compliance.
>
> __Q2: Can the Exploitation Stage Use a Non-Linear Oracle (e.g., Neural Network)?__
>
> __A2__. Yes. A core contribution of our IPGS design is its modularity. Non-linear methods (like neural networks) can be integrated seamlessly. The theoretical DBIC and regret structures still hold, though sample requirements scale with hypothesis class complexity.
>
> For non-linear oracles:
>
> - _The framework extends naturally_. Stage 1 (random diluted exploration) is model-agnostic. Stage 2 can use any oracle that provides reward predictions and uncertainty estimates.
> - _DBIC verification becomes harder_. Without closed-form confidence sets, verifying BIC requires bootstrap-based uncertainty quantification or conformal prediction sets.
> - _Regret guarantees change_. The $O(d\sqrt{T})$ bound relies on linear structure; with neural networks, regret depends on effective dimension or function class complexity.
>
> __Q3: Cold-Start Scales with K — Cost-Prohibitive for Large K?__
>
> __A3__: The cold-start sample complexity scales as $O(K^3 · d / \epsilon^2)$, which can indeed be large. We propose four practical mitigation strategies:
>
> - _Arm clustering_. Group similar arms and explore cluster representatives, reducing effective K. Natural in recommendation systems with feature-based item similarity.
> - _Progressive exploration_. Start with a reduced arm set (top-K by prior mean) and expand as the system learns.
> - _Warm starting_. Use historical data from previous users or A/B tests as informative priors, reducing cold-start samples needed.
> - _Contextual arm elimination_. Arms clearly suboptimal for a given context can be eliminated early, reducing effective K per context.
>
> __Q4: More Discussion of Strengths and Weakness__
>
> __1. Real-World User Behavior Assumption (Myopic, Rational, Stable Budget)__:
>
> We agree that real-world users exhibit adaptive and heterogeneous behavior. The assumption of myopic behavior is the standard foundational model in BIC literature (e.g., Mansour et al., 2020; Sellke & Slivkins, 2023) because it captures the fundamental principal-agent misalignment: users want the best product today, while the platform needs data for tomorrow.
>
> Our framework bridges this gap through the $\epsilon$-budget. Rather than assuming rigid rationality, $\epsilon$ models "bounded rationality" as a trust balance. Heterogeneous user rationality can be interpreted as personalized $\epsilon$ thresholds. While our analysis assumes stable $\epsilon$ for foundational bounds, platforms can dynamically adjust $\epsilon$ to accommodate adaptive behavior — an important direction for future empirical research.
>
> __2. On the Practical Validity and Robustness of Theoretical Assumptions__:
>
> - _Non-Degenerate Priors (Assumptions 1-2)_: Necessary to ensure the problem is not trivial. They guarantee a cold-start product has a "fighting chance"—its prior is not entirely stochastically dominated by safe arms, making exploration feasible.
>
> - _Evolving User Trust (Assumption 4)_: Models users becoming more open to platform signals over time. If violated (users remain stubborn), the algorithm remains robust but requires a longer Cold Start phase or looser incentive budget (see Appendix §F).
>
> - _Gaussian Priors_: Used primarily for clean, closed-form sample complexity bounds in Theorem 1. The core IPGS mechanism and general regret bounds extend to broader sub-Gaussian distributions, as noted in our formulation.

---

> > ### Author Rebuttal · Reviewer_ee1x · 2026-04-02
> >
> > All my concerns are resolved.

---

### Official Review · Reviewer_6D6j · 2026-03-10

**Soundness:** 3
**Presentation:** 3
**Significance:** 2
**Originality:** 3
**Overall Recommendation:** 4
**Confidence:** 3

**Summary:**

The paper studies the contextual bandit problem in recommender systems under the Bayesian Incentive Compatibility (BIC), i.e., the user has an incentive to accept or reject the recommendation (arm). The authors proposed a two-stage algorithm to address this problem. Specifically, in the first stage (cold start stage), they consider a random recommendation mechanism with a tuned probability to balance the exploration in uncertain arms while maintaining BIC. In the second stage, the authors employ an inverse proportional gap sampling algorithm. They show the theortical result of a sublinear regret bound under regularity assumptions.

**Compliance With Llm Reviewing Policy:**

Affirmed.

**Final Justification:**

I’ve read the rebuttal and will maintain my current acceptance score.

**Key Questions For Authors:**

1. Can the author discuss the necessity of a "two-stage" algorithm framework for this BIC context bandit problem that separates the cold start exploration and exploitation? Is that necessary in design due to the existence of the BIC?
2. I do not understand very clearly in lines 229–231 on page 5, i.e., why the pulls of the saturated arms are not counted in the history.
3. The authors take the regret of the cold start stage as the "price of incentives", but the second stage still needs to satisfy the BIC constraint. In general, how should I interpret the “price of incentives” here?

**Strengths And Weaknesses:**

Overall, the paper looks solid to me. I list a few concerns/comments below.

The paper assumes that users are myopic and strictly follow the dynamic BIC condition in Definition~1 as the behavioral rule for accepting or rejecting recommendations. It would be helpful for the paper to further justify this model choice & users' myopic behavior.

I'm not sure if I understand this correctly. It seems that under the authors' algorithm design, the user would never actually reject a recommendation, since the BIC constraint is enforced by the algorithm design at each stage. If this interpretation is correct, the main contribution of the paper is to develop such an algorithm for a linear contextual bandit problem subject to the dynamic BIC constraint in Definition~1. It would be helpful if the authors could more clearly highlight the novelty of their methodology in the context of the broader bandit literature.

Assumptions 1 & 2 are non-degeneracy conditions introduced for analytical convenience. But I'm not sure if they are well-justified under this specific problem.

Typos:
1. In 3.1, Notation paragraph, B_t = ..., missing a }
2. In Definition~1, I guess you mean i, j \in [K] and i\neq j?

---

> ### Author Rebuttal · Authors · 2026-03-28
>
> We thank the reviewer for the constructive feedback and for recognizing our paper solid. We address your comments below.
>
> __Q1: Justify the Myopic User Model Choice__
>
> __A1__: The myopic user assumption is standard in incentivized exploration literature (Mansour et al. 2015, 2020; Sellke & Slivkins 2024) and is motivated by practical considerations: In modern recommender systems, users make accept/reject decisions. The myopic model captures the empirically observed behavior that users evaluate recommendations based on their current belief about immediate reward. With $\epsilon=0$, the myopic user always selects the best content now; with $\epsilon>0$, they strategically accept content even if not currently optimal.
>
>
> __Q2: Users Never Reject — Clarify Novelty__
>
> __A2__: This is the key design. The algorithm ensures the DBIC constraint is always satisfied, meaning _rational_ users always accept, in both exploration and exploitation stages. The novelty lies in achieving this while maintaining near-optimal regret:
>
> - _A trivial DBIC-satisfying algorithm_ would always recommend the myopic-optimal arm — but this leads to _linear regret_ since no exploration occurs.
>
> - Our algorithm achieves two tradeoffs: a) _exploration vs. exploitation_ and b) _learning vs. incentive compatibility_. It explores sufficiently to learn (_sublinear regret_) while ensuring every recommendation satisfies DBIC (_no rejections_).
>
> You are correct: under our algorithm, a rational user never rejects because the algorithm strictly enforces the $\epsilon$ constraint at every step.
>
> Additionally, our methodology achieves this in a setting with _stochastic, dynamic_ user covariates, which precludes standard black-box reductions or posterior sampling methods used in prior literature. We solve this through our novel modular architecture using IPGS.
>
>
> __Q3: Justify Assumptions 1 and 2__
>
> __A3__:
> __Assumption 1__ (distinct prior means): Standard in BIC literature (Mansour et al., 2020). Ensures cold-start products have a "fighting chance"—their prior is not entirely dominated by saturated arms. Otherwise, incentivized exploration is trivially unnecessary.
>
> __Assumption 2__ (bounded context distribution): Ensures the optimal arm is eventually distinguishable (expected reward gap bounded away from zero). Naturally satisfied in recommendation systems where user features have bounded support (normalized features).
>
>
> __Q4: Is the Two-Stage Framework Necessary Due to BIC?__
>
> __A4__: Yes, the two-stage design is fundamentally driven by the learning characteristic coupled with BIC.
>
> _Stage 1 (cold start)_ is necessary because the platform has no data and the user's prior may strongly favor one arm. Pure exploitation yields no learning; pure exploration violates BIC. The diluted exploration mechanism (recommending myopic-best with probability $1-\alpha$, exploring with $\alpha$) is the only way to simultaneously satisfy BIC and collect initial samples. This sample complexity is the "price of incentives."
>
> _Stage 2_ becomes efficient because after Stage 1, the platform has enough data to form credible confidence intervals. The DBIC constraint becomes easier to satisfy—users trust "informed recommendations" since the mechanism demonstrably has relevant data. This enables more targeted exploration via inverse gap sampling.
>
> A single-stage approach would either (a) violate BIC in early rounds, or (b) use the conservative Stage 1 strategy throughout, leading to worse regret.
>
>
> __Q5: Why Are Saturated Arm Pulls Not Counted?__
>
> __A5__: Saturated arms are recommended as "organic" choices to subsidize exploration risk—this never loses user trust. By not counting them in exploration history, confidence sets in Stage 2 reflect only "useful" samples that reduce uncertainty, preventing over-counting. Saturated arms are still recommended (often as the best arm) but not tracked for epoch transitions.
>
>
> __Q6: Interpreting "Price of Incentives" When Stage 2 Also Satisfies BIC__
>
> __A6__: Great question! The "price of incentives" refers to the _additional regret_ in Stage 1 that _would not be necessary in a standard (non-IC) bandit algorithm_. In a standard contextual bandit, the learner can pull any arm directly—no cold-start cost. The Stage 1 cost ($T_{cold}(\epsilon)$) is entirely attributable to the BIC constraint.
>
> Stage 2 also satisfies BIC, but the constraint is essentially _"free"_ with our design—it does not increase regret beyond $O(d\sqrt{T})$. After Stage 1, BIC-compliant exploration coincides with statistically efficient exploration.
>
> In short, the "price" is paid upfront in Stage 1; after that, BIC comes for free.
>
>
> __Typos__: We will fix the typos noted.

---

> > ### Author Rebuttal · Reviewer_6D6j · 2026-04-03
> >
> > I would like to thank the authors for their detailed responses. I am inclined to maintain my current acceptance score.

---

### Official Review · Reviewer_CkgU · 2026-03-10

**Soundness:** 3
**Presentation:** 3
**Significance:** 3
**Originality:** 3
**Overall Recommendation:** 4
**Confidence:** 2

**Summary:**

The paper tackles the challenge of designing a recommendation system when the users are myopic and posses a certain prior on the environment. In this setting the incentives of the mechanisms and the ones of the users are misaligned, as the recommender is interested in exploring under explored options to learn knowledge, while users tend to avoid risky choices and would prefer to exploit the present knowledge.

To tackle this problem the paper proposes an online learning approach that respect the constraint of being $\epsilon$-Dynamic Bayesian Incentive-Compatible.

The proposed algorithm is composed of two parts. The first one focus on pure exploration, and to make it respect the DBIC constraint it dilutes the exploration with rounds of safer recommendations.

In the second phase it uses the acquired knowledge to provide a solid start on recommendations. It works on a basis of epochs of exponentially increasing length, and it adjust the parameter $\gamma_m$, so that as the epochs increase the recommendation concentrates to the predicted best arm, computed by an offline oracle.

**Compliance With Llm Reviewing Policy:**

Affirmed.

**Final Justification:**

I have read the responses and keep the original score

**Key Questions For Authors:**

--

**Limitations:**

yes

**Strengths And Weaknesses:**

The model seems interesting and well explained by possible applications, in particular it highlights the trade-off between gaining the trust of myopic self-interested users and maximize the long-term rewards.

The paper is well written and understandable, with an exposition which is sufficiently clear.

The backbone of the techniques are well highlighted and the basic intuition is quite easy to follow.  Unfortunately I am not familiar with the previous related literature, so it is hard for me to asses the true novelty of the techniques.

The paper is based on the exploitation of the information asymmetry, which is quite interesting. I appreciate that the authors have discussed the potential ethical problem related to this.

---

> ### Author Rebuttal · Authors · 2026-03-28
>
> We thank the reviewer for the positive feedback and for recognizing the importance of the trade-off between the user trust and exploration in modern recommender systems. And we also appreciate the positive assessment of our model, exposition, and ethical discussion.
>
> **Prior Literature**: To provide more context on our contributions compared to the existing literature, previous foundational works on BIC either focus on MAB without contexts (Mansour et al., 2020) or assume fixed-design linear contexts where arm features are static (Sellke and Slivkins, 2023). Our work is the first to establish $\epsilon$-DBIC in a dynamic setting with stochastic user covariates sampled online. Because the optimal arm changes per user, prior fixed-design analyses cannot be applied, which necessitated our novel two-stage framework and Inverse Proportional Gap Sampling (IPGS) approach. We will add a more prominent summary of Table 1 in the main text to help readers contextualize this novelty.
>
> **Planned Revision**:
> We plan to add a detailed **comparison table** in the related work section contrasting our contributions against prior BIC work (Kremer et al. 2014; Mansour et al. 2015, 2020; Sellke and Slivkins 2024) along key dimensions: context type, IC notion, regret formulation, and algorithmic approach.

---

> > ### Author Rebuttal · Reviewer_CkgU · 2026-04-04
> >
> > I appreciate the authors' efforts in providing a thorough rebuttal. I plan to keep my original score.

---

### Decision · Program_Chairs · 2026-04-30

**Decision:**

Accept (regular)

**Comment:**

The paper proposes the Recommendation Contextual Bandit (RCB) algorithm, which secures a $O(\sqrt{T})$ regret under the challenging situation of user noncompliance, where the user may not select the action recommended by the platform unless the recommended arm satisfies the $\epsilon$-DBIC constraint, where $\epsilon$ is the user’s incentive budget. RCB operates in two stages, where in the first stage, the platform balances between exploration (based on the prior) and organic recommendation (based on the posterior) so as to secure a sufficient number of samples for each arm so as to satisfy the $\epsilon$-DBIC constraint in the second stage. In the second stage, the platform then performs the standard inverse-proportional-gap-sampling, where the exploration constant gamma needs only be scaled with respect to the offline oracle’s generalization error bound and not to the $\epsilon$-DBIC constraint. The policies of the first and second stage are both tailored to the setting where user covariates are stochastic, a setting that has not been addressed in related literature. The second stage also differs from previous literature that employs posterior sampling, compatible with any offline regression oracle. Experiments on a real dataset confirm the efficiency of the algorithm, and authors analyze the decision error according to the budget value and strength of prior.

The reviewers agree that the paper is technically solid and addresses a problem that is both practically important and theoretically challenging. Their main request is that the authors strengthen the literature discussion, particularly regarding the choice of the myopic user model and the assumptions used in the paper. In particular, authors are recommended to clarify that these assumptions are standard in the existing literature and have practical relevance.